# Childhood trauma and schizotypy in non-clinical samples: A systematic review and meta-analysis

**Diamantis Toutountzidis**[1]*, **Tim M. Gale**[1,2], **Karen Irvine**[1], **Shivani Sharma**[1], **Keith R. Laws**[1]

**1** School of Life and Medical Sciences, University of Hertfordshire, Hatfield, United Kingdom, **2** Research and Development Department, Hertfordshire Partnership NHS Foundation Trust, Hatfield, United Kingdom

* d.toutountzidis@herts.ac.uk

**Data Availability Statement:** All relevant data are within the manuscript (the relevant correlations and sample sizes are in the paper itself). As this is a systematic review and meta-analysis, all the data

## Abstract

The association of early life adversities and psychosis symptoms is well documented in clinical populations; however, whether this relationship also extends into subclinical psychosis remains unclear. In particular, are early life adversities associated with increased levels of schizotypal personality traits in non-clinical samples? We conducted a systematic review and meta-analysis of associations between early life adversities and psychometrically defined schizotypal traits in non-clinical samples. The review followed PRISMA guidelines. The search using PubMed, Web of Science and EBSCO databases identified 1,609 articles in total. Twenty-five studies (N = 15,253 participants) met eligibility criteria for the review. An assessment of study quality showed that fewer than half of all studies were rated as methodologically robust. Meta-analyses showed that all forms of childhood abuse (emotional, physical and sexual) and neglect (emotional and physical) were significantly associated with psychometric schizotypy. The association of schizotypy traits with childhood emotional abuse (r = .33: 95%CI .30 to .37) was significantly larger than for all other form of abuse or neglect. Meta-regression analyses showed that the physical abuse-schizotypy relationship was stronger in samples with more women participants; and the sexual abuse-schizotypy relationship was stronger in younger samples. The current review identifies a dose-response relationship between all forms of abuse/neglect and schizotypy scores in non-clinical samples; however, a stronger association emerged for emotional abuse. More research is required to address the relationship of trauma types and specific symptom types. Future research should also address the under-representation of men.

## Introduction

Since the start of the last century, psychotic symptomatology has been viewed as ranging from the extremely severe through to milder clinical and sub-clinical presentations [1, 2]. Both Bleuler and Kraepelin observed that sub-clinical psychosis symptoms exist prior to onset of schizophrenia and may even be observed in biological relatives of patients. The modern expression

are derived from papers available in the public domain.

**Funding:** Part of the review was funded by a PhD bursary from the University of Hertfordshire to DT. The funder did not have any involvement in study design; in the collection, analysis and interpretation of data; in the writing of the report; and in the decision to submit the article for publication.

**Competing interests:** The authors have declared that no competing interests exist.

of this spectrum of experience is embodied in the continuum hypothesis of psychosis, which suggests that experiences such as hearing voices (auditory hallucinations) or delusional ideation occur in less distressing forms within the non-clinical population [3–5]. These experiences are not limited to the positive aspects of symptomatology (i.e., presence of odd feelings or behaviours), but extend also to negative symptoms (i.e., absence or lack of normal mental function) in non-clinical subjects [6].

Central to debates concerning the continuity of clinical, sub-clinical and healthy experiences has been the concept of schizotypy, which typically encompasses a broadly-defined phenotype of schizophrenia-like traits and behaviours that exist in the wider general population. The term "schizotypy" was first coined by Rado [7] and later adopted by Meehl [8], who viewed it as a form of latent personality organisation resulting from a specific biological/ genetic predisposition. Later work by Claridge [9] clarified the distinction between models that view schizotypy as dimensional or, as in Meehl's case, quasi-dimensional. Fully dimensional models would be those which suggest that schizotypy exists within both the healthy (and may even confer some advantages) and pathological range [10].

The continuum between healthy and pathological also encompasses other concepts, which have developed to describe *psychosis-like experiences* [PLEs; 11]. Schizotypy and PLEs refer to similar phenomena and both terms are frequently used interchangeably [12, 13] and reported as conceptually overlapping [14]. They might however, be distinguished in terms of the fact that PLEs are state-based and linked to symptoms (typically positive), while schizotypy is trait-based. Schizotypal traits are often classified into three domains that correspond to the key symptom areas of schizophrenia: the *positive* domain typically incorporates traits that relate to anomalies of cognition (e.g., paranoid ideation, ideas of reference); the *negative* domain includes interpersonal, emotional and deficit traits (e.g., anhedonia, no close friends); and the *disorganisation* domain includes traits related to disruptions in the ability to organise and express thoughts and behaviour (e.g. odd behaviour and odd speech). The extent to which various conceptions of schizotypy are related to the risk of schizophrenia is a matter of debate [15].

Amongst many risk factors, adverse childhood experiences have been linked to the experiences of psychosis [16], which might include various forms of physical abuse, verbal abuse, sexual abuse, physical neglect, and emotional neglect. Adversities in childhood (occurring before the age of 18) are common, with approximately 40% of the general adult population reporting at least one type of adverse experience (e.g., parental mental illness, domestic violence, physical, emotional, and sexual abuse, neglect) before the age of 18 [17]. Much of what is known about the impact of childhood trauma on adult mental health is gleaned from studies of clinical populations. Less is known about the impact of early life events on people who do not go on to receive a clinical psychosis diagnosis. By examining non-clinical samples expressing high and low schizotypal traits scores, we might shed light on the differences in types and/ or levels of trauma between clinical and non-clinical populations–and potentially increase understanding of the mechanisms through which trauma might lead to psychosis. One previous systematic review [18] of 25 studies reported that various types of childhood trauma were associated with schizotypy. The relationship of overall trauma with schizotypy revealed odds ratios across studies that ranged from 2.01 up to 4.15. The authors reported that associations exist for all trauma types, but concluded that it seemed stronger for emotional abuse. While they reported that trauma and schizotypy (especially positive traits) appeared to show a dose-response relationship, this relationship was based on a few individual studies rather than quantified and pooled across studies. Their review, which combined studies of both clinically-diagnosed schizotypal personality disorder (SPD) and those scoring high on self-rating schizotypy scales classified the majority of studies (14/25) as low in quality.

Since the review by Velikonja and colleagues [18], research in the relationship between childhood trauma and schizotypy in non-clinical samples has grown considerably, with several new large studies published. The aim of the current study is to systematically review the available published literature reporting associations between childhood traumatic experiences and psychometric schizotypal traits, as well as PLEs, in non-clinical samples alone. In particular, we will focus on the relationship between specific childhood trauma including: physical, emotional, and sexual abuse, as well as neglect, and psychometric schizotypy in non-clinical samples. Furthermore, we will conduct the first meta-analysis to quantify the reported associations between trauma types and schizotypy. Hereafter, all references to emotional, physical and sexual abuse, as well as physical and emotional neglect are to abuse and neglect experienced in childhood and up to the age of 18.

## Method

### Search strategy

Literature searches were conducted in three major databases (PubMed, EBSCO, and Web of Science) using the following sets of search terms:

> Child* abuse *OR* physical abuse *OR* sexual abuse *OR* psychological abuse *OR* emotional abuse *OR* child* trauma* *OR* child* advers* *OR* maltreat* *OR* bully* *OR* bullied *OR* stress* events *OR* early trauma* *OR* emotional neglect *OR* physical neglect

> *AND*

> Schizoty* *OR* psychosis-like *OR* psychotic-like *OR* magical ideation *OR* suspiciousness *OR* delusional ideation *OR* odd belie* *OR* eccentric behavi* *OR* odd speech *OR* constricted affect *OR* unusual perceptual experiences *OR* ideas of reference *OR* paranoid ideation

Publications between January 1980 and February 2021 were covered in all searches. The start-point was selected because the first investigations of the links between childhood trauma and psychosis outcomes appeared in the late 1980s, and because previous reviews focussing on clinical groups [e.g., 18, 19] failed to identify any relevant studies that took place prior to 1980. The Reference Manager EndNote X7 was used to manage the identified articles ($n = 2144$) and extract duplicates ($n = 535$). A total of 1609 articles were identified for title screening and 524 articles were found to be potentially relevant. Following abstract screening, 133 articles were identified for full-text screening and subsequently 25 articles were included in the present review (see S1 Appendix, for screened-excluded studies). All articles were independently screened by two authors (DT and KRL).

Two studies [i.e., 20, 21] employed the same sample. Only findings from Gong, Wang (20) are presented in this review, as they provided a more detailed account than Liu, Gong (21) on the links between different schizotypy dimensions and various types of childhood trauma. Wigman, van Winkel [22] and Kramer, Simons [23] used largely overlapping samples, and thus we report only findings from the latter, based on their relevance to the current review, as they provided results between childhood adversity and three different schizotypal outcomes–instead of one result between a schizotypal persistent group and overall childhood trauma that provided by the former. Sheinbaum, Racioppi [24] and Gong, Wang [25] used subsamples from Sheinbaum, Kwapil [26] and Gong, Wang [20] respectively, and thus only results from the latter studies were included in the current review. Although not preregistered, the review followed PRISMA guidelines for reporting systematic reviews and meta-analyses. A PRISMA

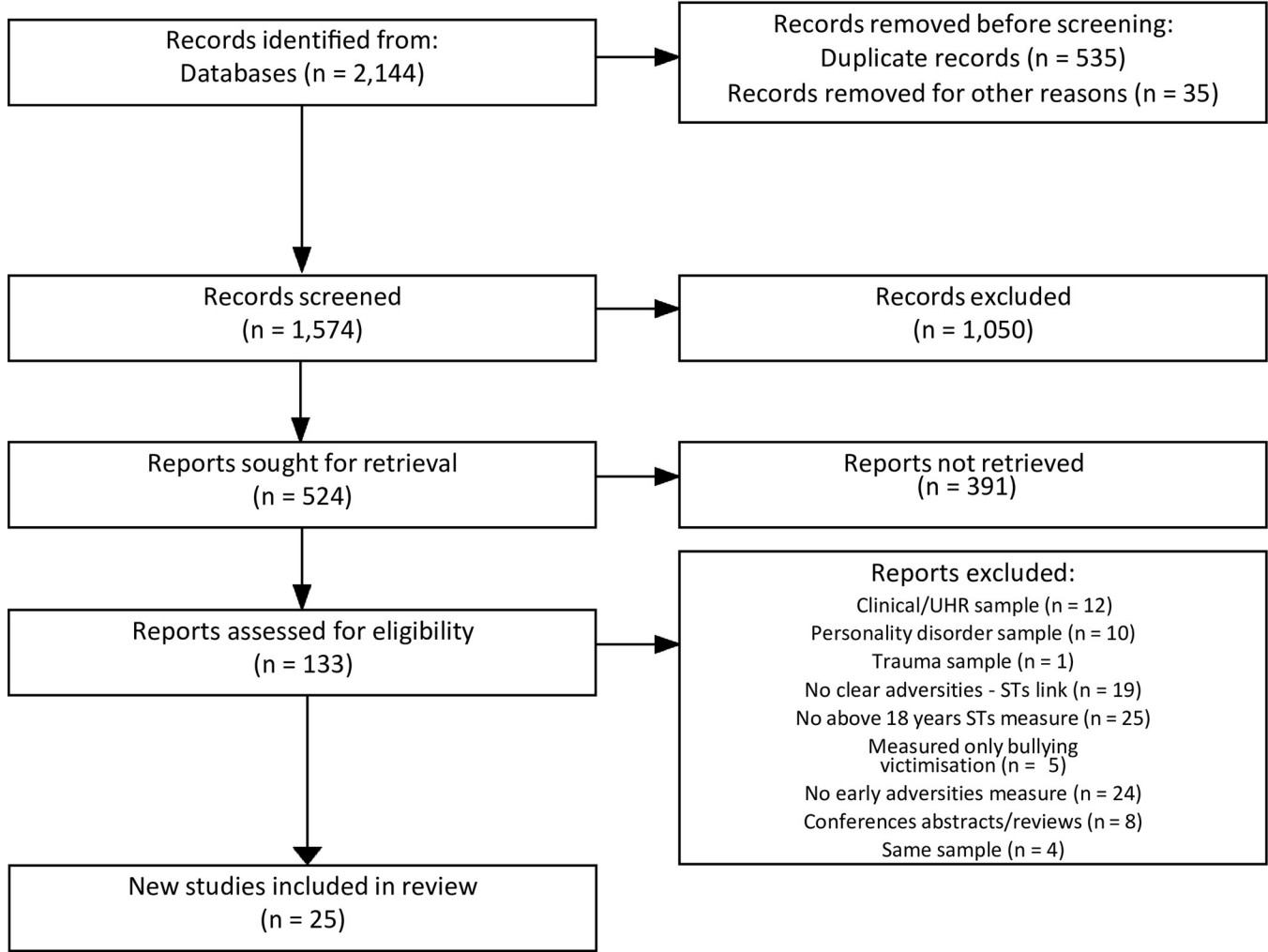

**Fig 1. PRISMA 2020 compliant flow diagram of each stage and details of excluded reports in full review.** STs = Schizotypal traits, UHR = Ultra-high-risk.

2020 compliant flow diagram tool [27] was used to provide record numbers of each stage and reasons for reports exclusion in the full review stage (Fig 1) and the checklist documents if and where relevant information may be located in the paper (see S2 Appendix, for the PRISMA 2020 Checklist).

### Eligibility criteria

Articles had to meet the following criteria to be included in the review: *(a)* be an original published research paper; *(b)* be written in English; *(c)* include a measure of childhood trauma for events before the age of 18 years (excluding peer victimisation; i.e., bullying; to assess for trauma where the perpetrator is an adult), *(d)* test for associations with schizotypy using any

standardised measure of either single symptom/trait or multidimensional schizotypal personality traits in adults (aged over 18); and *(e)* include general population samples and not clinical psychosis or personality disorder cases (though not necessarily screened for other personal or family mental health problems). Screening and eligibility assessment were performed independently by two reviewers (DT and KRL); and any disagreements were resolved by consensus.

## Quality assessment tool

Study quality was assessed using the quality assessment tool for childhood trauma and schizotypal traits research designed by Velikonja, Fisher [18]. Each study was rated on: sampling methods and sample sizes, selection and utilisation of standardised measures, and the assessment of confounding variables—scores from 0 to 2 were assigned on each item testing for the indicators. The maximum possible score for each study was 14 and the scorings (included in Table 3) were completed independently by two raters (DT and KRL), without disagreement.

## Corrected Covered Area (CCA)

We calculated the corrected covered area [CCA; 28] to determine the degree of overlap for primary studies included in the current review and the one previous systematic review of Velikonja, Fisher [18]. Pieper, Antoine [28] suggest that CCA scores between 0% and 5% represent 'slight overlap', 6%–10% 'moderate overlap', 11%–15% 'high overlap', and scores greater than 15% are considered to represent 'very high overlap'. CCA was calculated at 6.82% and so, represents slight-moderate overlap. This undoubtedly reflects the fact that 17 non-clinical studies have been published since Velikonja, Fisher [18] examined clinical and non-clinical samples.

## Meta-analysis of correlations

The statistical analysis was conducted using Comprehensive Meta-Analysis (version 2.2). All relevant data for calculating effect sizes (correlation values, sample sizes) and moderator variables (age, percentage of women participants and year of publication) were extracted independently by two authors (DT and KRL). Effect sizes were calculated for the correlations between schizotypy scores and each of the following: emotional abuse, physical abuse, sexual abuse, emotional neglect and physical neglect. All correlation coefficients were transformed to Fisher's z [29]. Synthesis of individual effect sizes to summary effect sizes was completed by conducting random effects meta-analyses. Results were then converted back from Fisher's z to $r$ for interpretation. Heterogeneity and variance among effect sizes of studies were assessed using the Q statistic and the $I^2$ statistic.

## Results

Following the full review stage, 25 articles were identified and included in the final analysis. The studies originated in the following national locations: Europe = 14; USA = 6; Australia = 2; Africa = 2; and China = 1. Quality assessment scores ranged from 4–10 out of a possible score of 14. The mean quality score was 7.32 (SD = 1.63); and only 11/25 (44%) of the studies met the cut-off score of 8+ used by Velikonja, Fisher [18] as a marker for the most methodologically robust studies. Study quality has remained consistent over time and shows no evidence of more recent improvement (see Fig 2).

## Sample

A total of 15,253 participants were included, with two-thirds being women ($n$ = 10,088; 66.1%); men ($n$ = 5,164; 33.9%). One participant self-identified their gender as 'other'. The

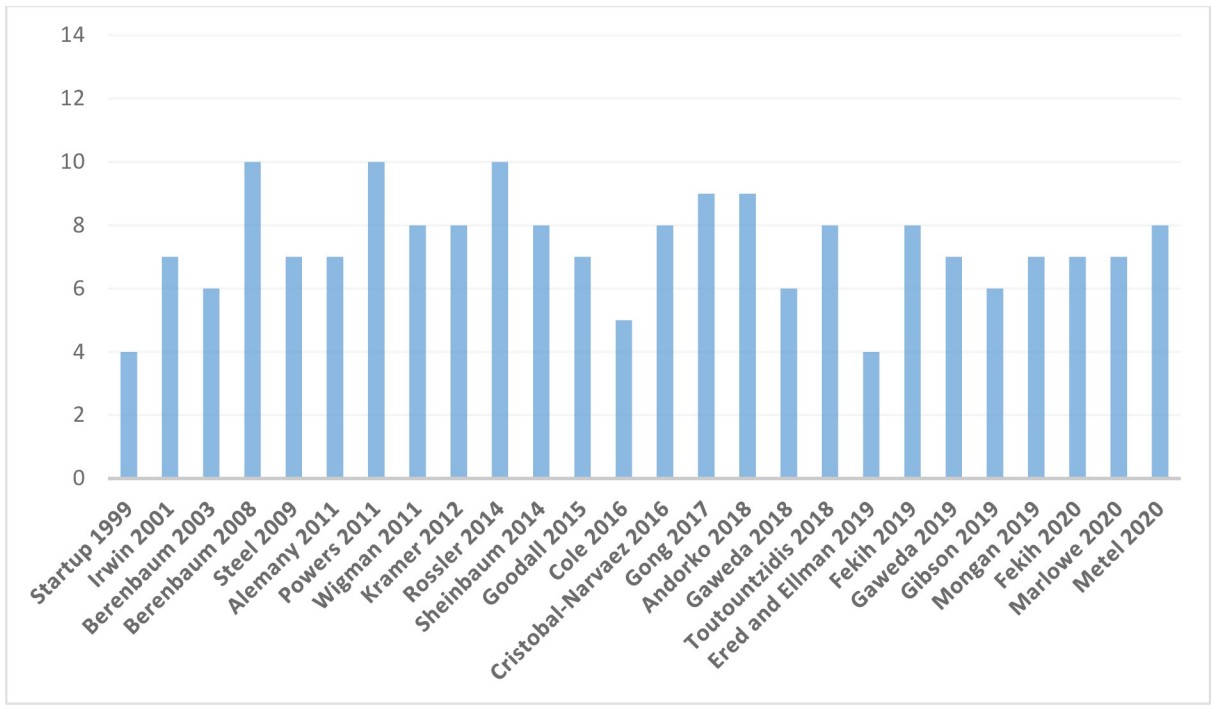

**Fig 2. Quality ratings for each study (maximum score = 14; cut-off for good quality = 8).**

mean age of participants across the 25 studies was 27.05 with a standard deviation of 12.36 (age range 18–95); median = 25.

## Standardised measures of childhood trauma

Various measures were used to assess childhood trauma, including: (a) Childhood Trauma Questionnaire, [CTQ; 30, 31]; (b) Early Trauma Inventory [ETI; 32, 33]; (c) Child Abuse and Trauma Scale [CATS; 34]; (d) Interview for Traumatic Events in Childhood [ITEC; 35]; (e) Traumatic Life Events Questionnaire [TLEQ; 36]; (f) Childhood Experience of Care and Abuse [CECA; 37]; (g) General Trauma Questionnaire [GTQ; as described in 38]; (h) Traumatic Events Checklist [TEC; 39]; (i) Structured Psychopathological Interview and Rating of the Social Consequences of Psychological Disturbances for Epidemiology [SPIKE; 40]; and (j) Adverse Childhood Experiences Questionnaire [ACE-Q; 41]. Both the CTQ and the ITEC assess five types of childhood traumatic events, including sexual, physical, and emotional abuse, and emotional neglect and physical neglect whilst growing up. The CATS, CECA and ACE-Q assess all aforementioned types of adversity as well as other adversities, such as parental conflict and control, parental mental illness, substance abuse, and family member going to prison. The ETI assesses physical, emotional and sexual abuse, as well as a range of general traumatic events, such as experience of natural disasters and serious personal injury of illness before the age of 18. The TLEQ assesses occurrence of childhood and adolescent physical and sexual abuse, as well as other types of trauma (e.g., natural disasters, accidents, death of close friends or relatives) throughout the lifespan. The GTQ covers four types of trauma (i.e., physical, sexual, emotional and general) that qualify as traumatic experiences in DSM-IV [42] A1 criteria for PTSD, as well as neglect. TEC captures emotional neglect, emotional, physical and sexual abuse, as well as other types of trauma, such as threatened death or serious injury. Eight items from the SPIKE capturing family problems, physical, emotional, and sexual abuse, were

**Table 1. Measures of early life adversity.**

| Adversity type / Scale | Physical abuse | Emotional abuse | Sexual abuse | Physical neglect | Emotional neglect | Other |
|---|---|---|---|---|---|---|
| ACE-Q[†] | X | X | X | X | X | X |
| CATS[§] | X | X | X | X | X | X |
| CECA[§] | X | X | X | X | X | X |
| CTQ[§] | X | X | X | X | X | |
| ETI[†] | X | X | X | | | X |
| GTQ[†] | X | | X | X | X | X |
| ITEC[†] | X | X | X | X | X | |
| SPIKE[†] | X | X | X | | | X |
| TEC[†] | X | X | X | | X | X |
| TLEQ[†] | X | | X | | | X |

*Note.* ACE-Q = Adverse Childhood Experience Questionnaire; CATS = Child Abuse and Trauma Scale; CECA = Childhood Experience of Care and Abuse; CTQ = Childhood Trauma Questionnaire; ETI = Early Trauma Inventory; GTQ = General Trauma Questionnaire; ITEC = Interview for Traumatic Events in Childhood; SPIKE = Structured Psychopathological Interview and Rating of the Social Consequences of Psychological Disturbances for Epidemiology; TEC = Traumatic Experiences Checklist; TLEQ = Traumatic Life Events Questionnaire. Examples of other include parental control, conflict with and between parents, accidents, death of family members and/or close friends, threat to life, experience of natural disasters.

Type of response

† = Yes/No

§ = Likert scale

also included. Table 1 includes adversity types examined by the different scales and type of response (e.g., yes/no or Likert). It is notable that the CTQ was the most commonly used approach to assessing childhood trauma–with 15/25 studies using the CTQ or one of its translated variants.

## Standardised measures of schizotypal experiences

Different measures of schizotypy were also utilised and these included: (a) Schizotypal Personality Questionnaire [SPQ; 43, 44]; (b) Community Assessment of Psychic Experiences [CAPE; 45]; (c) Schizotypal personality scale [STA; 46]; (d) Oxford-Liverpool Inventory of Feelings and Experiences [O-LIFE; 47]; (e) Structured Interview for Psychosis-risk Syndromes [SIPS; 48]; (f) Wisconsin Schizotypy Scales (WSS; including Perceptual Aberration [49]; Magical Ideation [50]; Physical Anhedonia [51]; and Revised Social Anhedonia [52] scales); (g) Launay-Slade Hallucination Scale-Revised [LSHS-R; 53]; (h) Peters et al. Delusional Inventory [PDI; 54, 55]; (i) the Prodromal Questionnaire [PQ; 56]; and (j) Five-Factor Schizotypal Inventory [FFSI; 57]. Most scales assess the positive, negative and disorganised domains of schizotypy that parallel the major symptom clusters of schizophrenia. The LSHS-R and PDI test for the single schizotypal traits of hallucination and delusional ideation respectively; STA that assesses positive schizotypy only; and WSS that does not assess disorganised traits. The PQ examines the presence/absence of attenuated symptoms of psychosis (i.e., symptoms which are below the level of severity that would warrant a diagnosis of a psychosis disorder) within the past four weeks; as well as any impact those symptoms may have on social functioning, academic/occupational functioning, and any related distress. In addition, three scales—Davos Assessment of Cognitive Biases [DACOBS; 58]; the Inventory of Psychotic-like Anomalous Self-Experiences [IPASE; 59]; and the Aberrant Salience Inventory [ASI; 60]—were included that capture cognitive biases, such as 'Jumping to Conclusions', and aberrant salience (i.e., the

**Table 2. Measures of schizotypy and various dimensions.**

| Traits / Scale | Positive | Negative | Disorganised | Hallucinations | Delusional Ideation | Cognitive biases | Associated distress |
|---|---|---|---|---|---|---|---|
| CAPE§ | X | X | | X | X | | X |
| DACOBS§ | | | | | | X | |
| FFSI§ | X | X | X | X | | | |
| IPASE§ | | | | X | | X | |
| LSHS§ | | | | X | | | |
| O-LIFE† | X | X | X | X | | | |
| PDI† | | | | | X | | X |
| PQ† | X | X | X | | | | X |
| SIPS† | X | X | X | X | X | | X |
| SPQ† | X | X | X | | | | |
| STA† | X | | | | | | |
| WSS† | X | X | | X | | | |

*Note*. CAPE = Community Assessment of Psychic Experiences; DACOBS = Davos Assessment of Cognitive Biases; FFSI = Five Factor Schizotypal Inventory; IPASE = Inventory of Psychotic-like Anomalous Self-Experiences; LSHS = Launay-Slade Hallucination Scale; O-LIFE = Oxford-Liverpool Inventory of Feelings and Experiences; PDI = Peters et al. Delusional Inventory; PQ = Prodromal Questionnaire; SIPS = Structured Interview for Psychosis-risk Syndromes; SPQ = Schizotypal Personality Questionnaire; STA = Schizotypal personality scale; WSS = Wisconsin Schizotypy Scales.

Type of response

† = Yes/No

§ = Likert scale

tendency to assign importance to inadequate stimuli). Domains examined by the various schizotypy scales, as well as type of response on each scale, are presented in Table 2.

## Association between childhood trauma and schizotypy

All studies (N = 25) reported at least one significant association between an early life adversity and some aspect of schizotypy. Emotional and physical abuse were associated with total schizotypy scores—with significant correlations ranging from $r = .25$ [61] to $r = .47$ [62] and from $r = .20$ [63] to $r = .40$ [62], respectively. Further, small-to-moderate significant associations between an array of psychosis-like symptomatology (i.e., unusual perceptions, delusional ideation, social anxiety, eccentric behaviour, constricted affect, eccentric behaviour, interpersonal and disorganised domains) and both emotional ($r = .10$ to .44) and physical abuse ($r = .10$ to .34) were observed across studies. For example, emotional abuse was positively associated with ideas of reference [64], aberrant salience [65], and hallucination-like experiences [66]. Examples of associations between physical abuse and specific schizotypal traits, included unusual perceptions [64] and delusional ideation [66].

Moderate associations were also observed between both physical and emotional neglect and different schizotypal dimensions [66, 67]. Similarly, positive associations were observed between sexual abuse and total schizotypy scores [61–63] as well as more specific traits, such as paranoia and suspiciousness [68], and unusual perceptual experiences [67, 68]. However, the associations with sexual abuse were smaller and less significant than the ones between emotional/physical abuse and schizotypy. Contrary to other studies that investigated the relationships of different types of childhood trauma and more than one dimensions of schizotypy, Steel, Marzillier [68] did not find an association with emotional abuse. Two studies reported associations in the aforementioned types of adversities and schizotypal traits for men and women separately [i.e., 61, 69]. Toutountzidis and colleagues found that physical abuse showed

links with an array of schizotypal traits only in women, that emotional abuse associated with an array of schizotypal traits in both genders and that sexual abuse did not have a link with schizotypy. However, Berenbaum and colleagues found links between various types of adversity (i.e., physical, emotional, and sexual abuse, and physical neglect) and schizotypy in both genders.

Cristobal-Narvaez, Sheinbaum [70] investigated links of total scores of abuse and neglect with psychosis-like, paranoia, having no thoughts or emotions, and negative affect domains. Overall, total scores of abuse and neglect were linked to psychosis-like paranoia, and negative affect domains, whereas no thoughts/emotions were linked only to neglect. Similarly, Alemany, Arias [71] examined links between both abuse and neglect with both positive and negative schizotypy dimensions. They reported significant links between childhood abuse and the positive dimension of schizotypy. Links between abuse and negative dimension of schizotypy, as well as neglect and both positive and negative dimensions were not found to be significant. Uniquely, significant bivariate generalised estimating equations of other types of childhood traumatic experiences (i.e., family problems, conflicts between parents, disliked, rejected, and repeated fights) on overall schizotypal signs were reported by Rössler, Hengartner [72]. Most studies employed cross-sectional designs–apart from Rössler, Hengartner [72] (prospective community study), and Kramer, Simons [23] (prospective twin study). Table 3 includes detailed results of all examined studies.

Overall, various types of childhood trauma (e.g., physical abuse, emotional abuse, sexual abuse, and neglect) and different schizotypy dimensions (e.g., positive, negative, disorganised) were observed to be associated across studies. This tripartite distinction was made based on the multidimensional nature of schizotypy–and by extension schizophrenia, with the positive dimension including disruptions in the content of thought and perceptual experiences (e.g., delusional ideation, hallucination, odd beliefs); negative dimension including diminished functioning (e.g., avolition, anhedonia); and disorganised dimension including disruptions in thought organisation and expression [123, 124]. Table 4 includes the numbers of times each of these trauma types were found to be significantly associated with schizotypy, as well as the times these associations were not observed to be significant. The numbers represent all findings within studies.

In terms of a simple vote count of reported associations, certain consistent associations emerge between trauma types and specific dimensions of schizotypy. Childhood emotional abuse invariably correlated with the negative dimension and with total schizotypy scores; Childhood neglect (physical and emotional) with the disorganised dimension; and finally, total childhood trauma scores with positive dimension scores.

## Meta-analysis

For those studies (in Table 3) reporting correlations between total schizotypy scores and key abuse and neglect measures: emotional abuse (k = 11), physical abuse (k = 13), sexual abuse (k = 12), emotional neglect (k = 7) and physical neglect (k = 9), data were pooled in a series of meta-analyses. Total schizotypy scores were selected as the outcome variable because insufficient numbers of studies provided data that could be used to examine schizotypal dimensions or more specific symptoms.

All correlation coefficients were transformed to Fisher's z. Synthesis of individual effect sizes to summary effect sizes was completed by conducting random effects meta-analyses; and results were then converted back from Fisher's z to $r$ for interpretation. Heterogeneity and variance among effect sizes of studies were assessed using Cochran's Q statistic and the $I^2$ statistic. Cochran's Q is calculated as the weighted sum of squared differences between individual study

**Table 3. Relevant studies on the associations between childhood trauma and schizotypy.**

| Author(s) (Year) | Quality score | Sample *N*, age range, Mean (SD) | Gender W/M | Measure of schizotypy | Measure of early life adversities | Other measures | Main findings in the association of early life adversities and psychosis-like symptoms | Types of adversity and domains of schizotypy analysed |
|---|---|---|---|---|---|---|---|---|
| Alemany, Arias [71] | 7 | 533, no range provided, 22.9 (5.4) | 291/ 242 | CAPE [45]; SPQ-B [44] | CTQ [31] | Cannabis use assessed with one question regarding the frequency of consumption; STAI [73] | Childhood abuse<br>Positive dimension ($\beta = .16^{**}$);<br>Negative dimension ($\beta = .11$)<br>Childhood neglect<br>Positive dimension ($\beta = -.09$);<br>Negative dimension ($\beta = -.03$) | Total scores of abuse/ neglect and different domains of schizotypy |
| Andorko, Millman [74] | 9 | 409, no range provided, 20.1 (3.22) | 207/202 | PQ-B [75] | GTQ-R [as described in 35] | ISDI [76]; BDI-II [77] | Total trauma ($r = .22^{***}$)<br>Combat ($r = —.07$)<br>Accident ($r = .15^{***}$)<br>Disaster ($r = .10^{**}$)<br>Witness ($r = .09$)<br>Raped ($r = .00$)<br>Sexually molested ($r = .08$)<br>Physical attack ($r = .09$)<br>Physical abuse ($r = .00$)<br>Serious neglect ($r = .06$)<br>Threatened with weapon ($r = .07$)<br>Other ($r = .12^{**}$)<br>Someone's else experience ($r = .22^{***}$) | Different types of early life adversities and total schizotypy scores |
| Berenbaum, Thompson [61]–Study 1 | 10 | 1510, 18–95, 44.2 (18.1) | 787/ 723 | SPQ [43] | Items adapted from instruments used in previous research measuring physical, emotional and sexual abuse, and physical neglect. | Threatening events were measured by asking participants whether they have experienced 11 events related to an injury or traumatic incidents (e.g., natural disaster, workplace injury). | Men:<br>Physical abuse ($r = .21^{**}$); Sexual abuse ($r = .12^{**}$); Emotional abuse ($r = .31^{**}$); Physical neglect ($r = .25^{**}$)<br>Women:<br>Physical abuse ($r = .24^{**}$); Sexual abuse ($r = .17^{**}$); Emotional abuse ($r = .25^{**}$); Physical neglect ($r = .25^{**}$) | Different types of early life adversities and total schizotypy scores |
| Berenbaum, Valera [62] | 6 | 75, 18–74, 38.7 (14.2) | 75/0 | SPQ [43]; Schizotypal personality disorder portion of SIDP-IV [78] | CTQ [30] | Included measures of PTSD, depression, dissociation, and alexithymia. | Physical abuse<br>SPQ total ($r = .40^{**}$); SPQ cognitive-perceptual ($r = .44^{**}$); SIDP total, ($r = .35^{**}$); SIDP cognitive-perceptual ($r = .36^{**}$)<br>Sexual abuse<br>SPQ total ($r = .32^{**}$); SPQ cognitive-perceptual ($r = .31^{**}$); SIDP total, ($.33^{**}$); SIDP cognitive-perceptual ($r = .30^{**}$)<br>Emotional abuse<br>SPQ total ($r = .47^{**}$); SPQ cognitive-perceptual ($r = .43^{**}$); SIDP total ($r = .36^{**}$); SIDP cognitive-perceptual ($r = .36^{**}$)<br>Physical neglect<br>SPQ total ($r = .60^{**}$); SPQ cognitive-perceptual ($r = .51^{**}$); SIDP total ($r = .34^{**}$); SIDP cognitive-perceptual ($r = .42^{**}$)<br>Emotional neglect<br>SPQ total ($r = .56^{**}$); SPQ cognitive-perceptual ($r = .43^{**}$); SIDP total ($r = .54^{**}$); SIDP cognitive-perceptual ($r = .49^{**}$) | Different types of early life adversities and total schizotypy scores |

*(Continued)*

**Table 3.** (*Continued*)

| Author(s) (Year) | Quality score | Sample *N*, age range, Mean (SD) | Gender W/M | Measure of schizotypy | Measure of early life adversities | Other measures | Main findings in the association of early life adversities and psychosis-like symptoms | Types of adversity and domains of schizotypy analysed |
|---|---|---|---|---|---|---|---|---|
| Cole, Newman-Taylor [66] | 5 | 200, 18–38, 19.96 (2.18) | 165/34 (one as other) | LSHS-R [50]; PDI-21 [52] | CATS [31] | DES-II [79]; CDS [80] | Sexual abuse<br>LSHS-R ($r_s$ = .20**)<br>PDI ($r_s$ = .12)<br>Physical abuse<br>LSHS-R ($r_s$ = .27**)<br>PDI ($r_s$ = .28**)<br>Emotional abuse<br>LSHS-R ($r_s$ = .43**)<br>PDI ($r_s$ = .36**)<br>Neglect<br>LSHS-R ($r_s$ = .44**)<br>PDI ($r_s$ = .46**) | Different types of early life adversities and different domains of schizotypy |
| Cristobal-Narvaez, Sheinbaum [70] | 8 | 206, no range provided, 21.3 (2.4) | 162/44 | Experience sampling methodology measuring indices of paranoia, psychosis-like symptoms, no thoughts/emotions negative affect | CTQ [31]; ITEC [35]; General trauma subscale of the ETI [32] | Bullying by peers was assessed with questions from the CECA [37] | Abuse CTQ<br>Psychosis-like index ($\beta$ = .009**)<br>Paranoia index ($\beta$ = .022**); No thoughts/emotions ($\beta$ = .002)<br>Negative affect ($\beta$ = .035***)<br>Neglect CTQ<br>Psychosis-like index ($\beta$ = .009**);<br>Paranoia index ($\beta$ = .023**); No thoughts/emotions ($\beta$ = .10*);<br>Negative affect ($\beta$ = .027**)<br>Abuse ITEC<br>Psychosis-like index ($\beta$ = .007**);<br>Paranoia index ($\beta$ = .016***); No thoughts/emotions ($\beta$ = .007);<br>Negative affect ($\beta$ = .024***)<br>Neglect ITEC<br>Psychosis-like index ($\beta$ = .006*);<br>Paranoia index ($\beta$ = .013*); No thoughts/emotions ($\beta$ = .009)<br>Negative affect ($\beta$ = .018*) | Total scores of abuse/ neglect and different domains of schizotypy |
| Ered and Ellman [81] | 4 | 130, no range provided, 19.68 (1.9) | 104/26 | 45 positive items of PQ [56]; SIPS [48] | CTQ [30] | | Emotional abuse<br>Unusual Thought Content ($r$ = .00); Paranoid Ideation ($r$ = .21*); Grandiosity ($r$ = .00); Perceptual Disturbances ($r$ = .12); Disorganisation ($r$ = .27***)<br>Physical abuse<br>Unusual Thought Content ($r$ = .07); Paranoid Ideation ($r$ = .12); Grandiosity ($r$ = .00); Perceptual Disturbances ($r$ = .03); Disorganisation ($r$ = .19*)<br>Sexual abuse<br>Unusual Thought Content ($r$ = -.03); Paranoid Ideation ($r$ = .05); Grandiosity ($r$ = -.03); Perceptual Disturbances ($r$ = .01); Disorganisation ($r$ = .28**)<br>Emotional neglect<br>Unusual Thought Content ($r$ = .12); Paranoid Ideation ($r$ = .32**); Grandiosity ($r$ = -.04); Perceptual Disturbances ($r$ = .21*); Disorganisation ($r$ = .27**)<br>Physical neglect<br>Unusual Thought Content ($r$ = .07); Paranoid Ideation ($r$ = .11); Grandiosity ($r$ = -.01); Perceptual Disturbances ($r$ = .19*); Disorganisation ($r$ = .25**) | Different types of early life adversities and different domains of schizotypy |

(*Continued*)

**Table 3.** (*Continued*)

| Author(s) (Year) | Quality score | Sample *N*, age range, Mean (SD) | Gender W/M | Measure of schizotypy | Measure of early life adversities | Other measures | Main findings in the association of early life adversities and psychosis-like symptoms | Types of adversity and domains of schizotypy analysed |
|---|---|---|---|---|---|---|---|---|
| Fekih-Romdhane, Nsibi [82] | 7 | 75, no range provided, 23.4 (2.1) | 37 / 38 | CAPE French version [83] | CTQ French version [84] | DASS-21 French version [85] | Emotional abuse<br>Positive dimension ($r = .41^{**}$);<br>Negative dimension ($r = .37^{**}$);<br>Depressive dimension ($r = .34^{**}$)<br>Physical abuse<br>Positive dimension ($r = .21$);<br>Negative dimension ($r = .14$)<br>Depressive dimension ($r = .16$)<br>Sexual abuse<br>Positive dimension ($r = .53^{**}$);<br>Negative dimension ($r = .34^{**}$);<br>Depressive dimension ($r = .32^{**}$)<br>Emotional neglect<br>Positive dimension ($r = .33^{**}$)<br>Negative dimension ($r = .25^{*}$);<br>Depressive dimension ($r = .31^{**}$)<br>Physical neglect<br>Positive dimension ($r = .42^{**}$);<br>Negative dimension ($.32^{**}$);<br>Depressive dimension ($r = .33^{**}$) | Different types of early life adversities and different domains of schizotypy |
| Fekih-Romdhane, Tira [86] | 8 | 482 18–32, 22.1 (2.7) | 307/ 175 | CAPE French version [83] | CTQ French version [84] | DASS-21 French version [85] | Emotional abuse<br>Positive dimension ($r = .29^{**}$);<br>Negative dimension ($r = .40^{**}$);<br>Depressive dimension ($r = .44^{**}$)<br>Physical abuse<br>Positive dimension ($r = .30^{**}$);<br>Negative dimension ($r = .21^{**}$);<br>Depressive dimension ($r = .24^{**}$)<br>Sexual abuse<br>Positive dimension ($r = .42^{**}$);<br>Negative dimension ($r = .30^{**}$);<br>Depressive dimension ($r = .28^{**}$)<br>Emotional neglect<br>Positive dimension ($r = .12^{*}$)<br>Negative dimension ($r = .08$);<br>Depressive dimension ($r = .08$)<br>Physical neglect<br>Positive dimension ($r = .09^{*}$);<br>Negative dimension ($r = .00$);<br>Depressive dimension ($r = .02$) | Different types of early life adversities and different domains of schizotypy |
| Gaweda, Goritz [65] | 7 | 649, 21–80, 51.1 (14) | 358/ 291 | PQ-16 [87]; ASI [60]; IPASE [59] | CTQ [30] | | Emotional abuse<br>PQ-16 ($r = .30^{***}$); ASI ($r = .24^{***}$); IPASE ($r = .30^{***}$)<br>Physical abuse<br>PQ-16 ($r = .23^{***}$); ASI ($r = .15^{***}$); IPASE ($r = .28^{***}$)<br>Sexual abuse<br>PQ-16 ($r = .18^{***}$); ASI ($r = .17^{***}$); IPASE ($r = .20^{***}$)<br>Emotional neglect<br>PQ-16 ($r = .14^{***}$); ASI ($r = .09^{*}$); IPASE ($r = .32^{**}$)<br>Physical neglect<br>PQ-16 ($r = .17^{***}$); ASI ($r = .11^{**}$); IPASE ($r = .34^{***}$) | Different types of early life adversities and total schizotypy scores |

(*Continued*)

**Table 3.** (Continued)

| Author(s) (Year) | Quality score | Sample N, age range, Mean (SD) | Gender W/M | Measure of schizotypy | Measure of early life adversities | Other measures | Main findings in the association of early life adversities and psychosis-like symptoms | Types of adversity and domains of schizotypy analysed |
|---|---|---|---|---|---|---|---|---|
| Gaweda, Prochwicz [88] | 6 | 653, 18–37, 22.24 (3.14) | 463/190 | CAPE [45]; IPASE [59]; DACOBS [58] | TEC [39] | | TEC total <br> Cognition ($r = .15^{***}$); Somatisation ($r = .20^{***}$); Demarcation ($r = .16^{***}$); Consciousness ($r = .18^{***}$); Self-awareness ($r = .19^{***}$); IPASE total ($r = .20^{***}$); Beliefs inflexibility ($r = .07$); Jumping to Conclusions ($r = .02$); Attention to threat ($r = .15^{***}$); External Attributions ($r = .30^{***}$); CAPE total ($r = .28^{***}$); Positive symptoms ($r = .27^{***}$); Negative symptoms ($r = .21^{***}$) | Total trauma and positive and negative traits of schizotypy |
| Gibson, Reeves [89] | 6 | 945, 18–34, 20.31 (2.47) | 714/231 | 45 positive items of PQ [56] | CTQ [31] | PSS [90]; BCSS [91]; DES [92]; Rotter I-E [93]; CES-D [94]; STAI [73, 95]; DUF [96] | Total trauma <br> ($b = .16^{***}$) | Total scores of adversity and total schizotypy |
| Gong, Wang [20] | 9 | 2469, no range provided, Women: 18.74 (1.14) men: 18.79 (1.09) | 1785/684 | Chinese version of the SPQ [97] | Chinese version of the CTQ [98] | Chinese version of AQ [99] | Emotional abuse <br> SPQ total ($r = .38^{**}$); Positive ($r = .34^{**}$); Negative ($r = .35^{**}$); Disorganised ($r = .32^{**}$) <br> Physical abuse <br> SPQ total ($r = .24^{**}$); Positive ($r = .22^{**}$); Negative ($r = .23^{**}$); Disorganised ($r = .19^{**}$) <br> Sexual abuse <br> SPQ total ($r = .27^{**}$); Positive ($r = .27^{**}$); Negative ($r = .25^{**}$); Disorganised ($r = .19^{**}$) <br> Emotional neglect <br> SPQ total ($r = .20^{**}$); Positive ($r = .12^{**}$); Negative ($r = .24^{**}$); Disorganised ($r = .17^{**}$) <br> Physical neglect <br> SPQ total ($r = .24^{**}$); Positive ($r = .20^{**}$); Negative ($r = .24^{**}$); Disorganised ($r = .20^{**}$) | Different types of early life adversities and different domains of schizotypy |
| Goodall, Rush [63] | 7 | 283, 18–74, 26.8 (9.28) | 203/80 | SPQ-B [44] | CTQ [31] | ECR-R [100] | Emotional abuse ($r_s = .42^{**}$); Emotional neglect ($r_s = .30^{**}$); Physical abuse ($r_s = .20^{**}$); Physical neglect ($r_s = .33^{**}$); Sexual abuse ($r_s = .13^{*}$) | Different types of early life adversities and total schizotypy scores |
| Irwin [67] | 7 | 116, 18–46, 22.7 (7.36) | 74/42 | SPQ-B [44] | CTQ [30] | DES [92] | Physical and emotional abuse <br> Cognitive-Perceptual ($r = .46^{***}$); Interpersonal ($r = .28^{**}$); Disorganised ($r = .39^{***}$) <br> Emotional neglect <br> Cognitive-Perceptual ($r = .36^{***}$); Interpersonal ($r = .28^{**}$); Disorganised ($r = .32^{***}$) <br> Physical neglect <br> Cognitive-Perceptual ($r = .34^{***}$); Interpersonal ($r = .26^{**}$); Disorganised $r = .31^{***}$) <br> Sexual abuse <br> Cognitive-Perceptual ($r = .21^{*}$); Interpersonal ($r = .08$); Disorganized ($r = .20^{*}$) | Different types of early life adversities and different domains of schizotypy |

*(Continued)*

**Table 3.** (*Continued*)

| Author(s) (Year) | Quality score | Sample *N*, age range, Mean (SD) | Gender W/M | Measure of schizotypy | Measure of early life adversities | Other measures | Main findings in the association of early life adversities and psychosis-like symptoms | Types of adversity and domains of schizotypy analysed |
|---|---|---|---|---|---|---|---|---|
| Kramer, Simons [23] | 8 | 508, 18–46, 27.1 (7.4) | 508 / 0 | CAPE [45] Paranoid ideation and psychoticism of SCL-90-R [101] Delusions and hallucinations subscales of SCID-I [102] | Dutch translation of the original 70-item CTQ [30, 103] | Depressive symptoms were measured using depression subscales of SCL-90-R and SCID-I Stress sensitivity was measured in daily life using the Experience Sampling Method | Total trauma CAPE ($\beta = .13^{***}$) SCL-90-R ($\beta = .16^{***}$) SCID ($\beta = .09$) | Total scores of childhood adversity and total schizotypy |
| Marlowe, Perry [104] | 7 | 298, 18–64, 33.08 (10.65) | 223 / 75 | CAPE [45] | CTQ-Brief [105] | OIS-34 [106] PAM [107] PBI [108] TAPS-1 [109] MHHQ [106] | Emotional abuse Positive dimension ($r = .24^*$); Negative dimension ($r = .22^*$); Depressive dimension ($r = .28^*$) Physical abuse Positive dimension ($r = .18^*$); Negative dimension ($r = .20^*$); Depressive dimension ($r = .16^*$) Sexual abuse Positive dimension ($r = .11$); Negative dimension ($r = .03$); Depressive dimension ($r = .07$) Emotional neglect Positive dimension ($r = .17^*$) Negative dimension ($r = .18^*$); Depressive dimension ($r = .17^*$) Physical neglect Positive dimension ($r = .18^*$); Negative dimension (.10); Depressive dimension ($r = .14^*$) | Different types of early life adversities and different domains of schizotypy |
| Metel, Arciszewska [110] | 8 | 2614, 18–35, 26.37 (4.71) | 1673/ 941 | PQ-16 [87]; DACOBS [58] | TEC [39]; CECA [37] | CD-RISC 10 [111]; CES-D [94] | Total trauma PQ ($r = .34^{**}$); DACOBS ($r = .30^{**}$) | Total scores of adversity and total schizotypy |
| Mongan, Shannon [112] | 7 | 748, 18–35, 27.93 (4.34) | 331 / 417 | PQ-16 [86] | ACE-Q [38] | Brief COPE [113] MSPSS [114] BRS [115] NCS [116] | Verbal abuse/threat ($\beta = .15^{***}$) Sexual abuse ($\beta = .10^{**}$) Emotional neglect ($\beta = .11^{**}$) Physical neglect ($\beta = .15^{***}$) Household mental health difficulties ($\beta = .09^*$) | Different types of early life adversities and total schizotypy scores |

(*Continued*)

**Table 3.** (Continued)

| Author(s) (Year) | Quality score | Sample N, age range, Mean (SD) | Gender W/M | Measure of schizotypy | Measure of early life adversities | Other measures | Main findings in the association of early life adversities and psychosis-like symptoms | Types of adversity and domains of schizotypy analysed |
|---|---|---|---|---|---|---|---|---|
| Powers, Thomas [64] | 10 | 541, Median age of 41 | 319/222 | Schizotypal measures of SNAP [117] i.e., Ideas of Reference; Odd Beliefs; Unusual Perceptions; Eccentric Behaviour; Constricted Affect; Social Anxiety; Lack of Close Friends; Suspiciousness | CTQ [30]; ETI [32] | CAPS [118] | CTQ and SNAP links<br>Physical abuse<br>Ideas of Reference ($r$ = .07); Odd Beliefs ($r$ = .04); Unusual perceptions ($r$ = .11**); Eccentric Behaviour ($r$ = .15***); Constricted Affect ($r$ = .12***); Social Anxiety ($r$ = .06); Lack of Close Friends ($r$ = .10*); Suspiciousness ($r$ = .08)<br>Emotional abuse<br>Ideas of Reference ($r$ = .19***); Odd Beliefs ($r$ = .10*); Unusual perceptions ($r$ = .15***); Eccentric Behaviour ($r$ = .27***); Constricted Affect ($r$ = .17***); Social Anxiety ($r$ = .16***); Lack of Close Friends ($r$ = .21***); Suspiciousness ($r$ = .12**)<br>Sexual abuse<br>Ideas of Reference ($r$ = .07); Odd Beliefs ($r$ = .05); Unusual perceptions ($r$ = .08); Eccentric Behaviour ($r$ = .11*); Constricted Affect ($r$ = .04); Social Anxiety ($r$ = .04); Lack of Close Friends ($r$ = .06); Suspiciousness ($r$ = .05)<br>ETI and SNAP links<br>Physical abuse<br>Ideas of Reference ($r$ = .09); Odd Beliefs ($r$ = -.03); Unusual perceptions ($r$ = .10*); Eccentric Behaviour ($r$ = .17***); Constricted Affect ($r$ = .14**); Social Anxiety ($r$ = .12*); Lack of Close Friends ($r$ = .08); Suspiciousness ($r$ = .00)<br>Emotional abuse<br>Ideas of Reference ($r$ = .15**); Odd Beliefs ($r$ = .04); Unusual perceptions ($r$ = .17***); Eccentric Behaviour ($r$ = .23***); Constricted Affect ($r$ = .10*); Social Anxiety ($r$ = .17***); Lack of Close Friends ($r$ = .16***); Suspiciousness ($r$ = .07)<br>Sexual abuse<br>Ideas of Reference ($r$ = .07); Odd Beliefs ($r$ = .01); Unusual perceptions ($r$ = .07); Eccentric Behaviour ($r$ = .07); Constricted Affect ($r$ = -.02); Social Anxiety ($r$ = .05); Lack of Close Friends ($r$ = .01); Suspiciousness ($r$ = .00) | Different types of early life adversities and different domains of schizotypy |

(*Continued*)

**Table 3.** (*Continued*)

| Author(s) (Year) | Quality score | Sample *N*, age range, Mean (SD) | Gender W/M | Measure of schizotypy | Measure of early life adversities | Other measures | Main findings in the association of early life adversities and psychosis-like symptoms | Types of adversity and domains of schizotypy analysed |
|---|---|---|---|---|---|---|---|---|
| Rössler, Hengartner [72] | 10 | 335, assessed between 1978 (around age 20) and 2008 (around age 50). | 191/144 | Two symptom dimensions relevant for psychosis ("paranoid ideation" and "psychoticism") from the SCL-90-R [101] | SPIKE [40] | | Bivariate generalised estimating equations of childhood adversities with overall schizotypal signs. Broken home ($\beta$ = .02); Family problems ($\beta$ = .04*); Conflicts between parents ($\beta$ = .06**); Conflicts with parents ($\beta$ = .02); Sexual abuse ($\beta$ = .06); Severe punishment ($\beta$ = .03); Disliked, rejected ($\beta$ = .057*); Repeated fights ($\beta$ = .080*); Total adversity score ($\beta$ = .02**) | Different types of early life adversities and total schizotypy scores |
| Sheinbaum, Kwapil [26] | 8 | 546, no range provided, 20.6 (4.1) | 454/92 | Positive symptom subscale of CAPE [45]; Paranoid beliefs with the suspiciousness subscale of the SPQ [43]; WSS [49, 50, 51, 52] | CTQ [31] | Attachment style measured with the RQ [119] | Physical/Emotional trauma<br>Psychosis-like experiences ($r$ = .22***); Suspiciousness ($r$ = .27***); Positive schizotypy ($r$ = .22***); Negative schizotypy ($r$ = .25***)<br>Sexual abuse<br>Psychosis-like experiences ($r$ = .07); Suspiciousness ($r$ = .02); Positive schizotypy ($r$ = .09); Negative schizotypy ($r$ = -.02) | Different types of early life adversities and different domains of schizotypy |
| Startup [120] | 4 | 224, no range provided, 39.1 (18.5) | 144/80 | O-LIFE [47]; testing for Unusual experiences, Cognitive disorganization, Introvertive anhedonia | Two questions previously employed by Bryer, Nelson [121] assessed sexual and physical abuse | DES [79]; Lie scale of the EPQ [122] | Physical abuse<br>Unusual Experiences ($r$ = .11); Cognitive Disorganization ($r$ = .07); Introvertive anhedonia ($r$ = .02)<br>Sexual abuse<br>Unusual Experiences ($r$ = .14*); Cognitive Disorganization ($r$ = .06); Introvertive anhedonia ($r$ = —.10) | Different types of early life adversities and different domains of schizotypy |
| Steel, Marzillier [68] | 7 | 384, 18–67, 24.9 (7.2) | 292/92 | STA [46] | TLEQ [36] One question from the CTQ [30] to capture emotional abuse and neglect | DASS-21 [85]; BCSS [91] | Physical abuse<br>Magical thinking (OR 4.49, 95% CI 1.0–21.0); Paranoia and suspiciousness (OR 5.84*, 95% CI 1.5–23.1); Unusual perceptual experiences (OR 6.46*, 95% CI 1.1–36.9)<br>Emotional abuse<br>Magical thinking (OR 1.55, 95% CI 0.6–4.2); Paranoia and suspiciousness (OR 2.45, 95% CI 0.9–6.8); Unusual perceptual experiences (OR 2.61, 95% CI 0.6–11.5)<br>Sexual abuse<br>Magical thinking (OR 1.31, 95% CI 0.4–4.0); Paranoia and suspiciousness (OR 4.49**, 95% CI 1.7–12.2); Unusual perceptual experiences (OR 4.00*, 95% CI 1.0–15.2) | Different types of early life adversities and different domains of schizotypy |

**Table 3.** (Continued)

| Author(s) (Year) | Quality score | Sample *N*, age range, Mean (SD) | Gender W/M | Measure of schizotypy | Measure of early life adversities | Other measures | Main findings in the association of early life adversities and psychosis-like symptoms | Types of adversity and domains of schizotypy analysed |
|---|---|---|---|---|---|---|---|---|
| Toutountzidis, Gale [69] | 8 | 320, 18–75, 28.24 (12.76) | 221/99 | FFSI [57] | Physical, Emotional and Sexual abuse scales of the ETI-SF [33] | | Men: <br> Physical abuse <br> (*none significant*) <br> Emotional abuse <br> Interpersonal Suspiciousness, ($r_s = .39^{***}$); Social Anhedonia ($r_s = .37^{***}$); Social Isolation ($r_s = .30^{**}$); Physical Anhedonia ($r_s = .28^{**}$); Social Anxiousness ($r_s = .30^{**}$); Social Discomfort ($r_s = .28^{**}$); Aberrant Perceptions ($r_s = .33^{***}$); <br> Sexual abuse <br> (*none significant*) <br> women: <br> Physical abuse <br> Interpersonal Suspiciousness ($r_s = .24^{***}$); Social Anhedonia ($r_s = .18^{**}$); Social Isolation ($r_s = .26^{**}$); Physical Anhedonia ($r_s = .23^{***}$); Social Anxiousness ($r_s = .19^{**}$); Odd & Eccentric ($r_s = .27^{***}$); Aberrant Ideas ($r_s = .28^{***}$); Aberrant Perceptions ($r_s = .34^{***}$) <br> Emotional abuse <br> Interpersonal Suspiciousness, ($r_s = .31^{***}$); Social Anhedonia ($r_s = .32^{***}$); Social Isolation ($r_s = .31^{***}$); Physical Anhedonia ($r_s = .35^{***}$); Social Anxiousness ($r_s = .29^{***}$); Social Discomfort ($r_s = .26^{**}$); Odd & Eccentric;, ($r_s = .29^{***}$); Aberrant Ideas ($r_s = .30^{***}$); Aberrant Perceptions ($r_s = .29^{***}$) <br> Sexual abuse <br> Aberrant Ideas ($r_s = .18^{**}$; Aberrant Perceptions ($r_s = .22^{***}$) | Different types of early life adversities and different domains of schizotypy |

*Note.* ACE-Q = Adverse Childhood Experiences Questionnaire; AQ = Autism-Spectrum Quotient; ASI = Aberrant Salience Inventory; BCSS = Brief Core Schema Scale; BDI-II = Beck Depression Inventory—Second Edition; BRS = Brief Resilience Scale; CAPE = Community Assessment of Psychic Experiences; CAPS = Clinician-Administered PTSD Scale; CATS = Child Abuse and Trauma Scale; CD-RISC 10 = Connor-Davidson Resilience Scale; CES-D = Center for Epidemiologic Studies Depression Scale; CDS = Cambridge Depersonalization Scale; CECA = Childhood Experience of Care and Abuse; CTQ = Childhood Trauma Questionnaire; DACOBS = Davos Assessment of Cognitive Biases; DASS-21 = Depression Anxiety Stress Scales—21 items; DES-II = Dissociative Experiences Scale-II; DUF = Drug Use Frequency; ECR-R = Experiences in Close Relationships Questionnaire-Revised; EPQ = Eysenck Personality Questionnaire; ETI-SF = Early Trauma Inventory-Short Form; FFSI = Five Factor Schizotypal Inventory; GTQ-R = General Trauma Questionnaire-Revised; IPASE = Inventory of Psychotic-like Anomalous Self-Experiences; ISDI = Iowa Sleep Disturbances Inventory; ITEC = Interview for Traumatic Events in Childhood; LSHS-R = Launay-Slade Hallucination Scale-Revised; MHHQ = Mental Health History Questionnaire; MSPSS = Multidimensional Scale of Perceived Social Support; NCS = Neighbourhood Cohesion Scale; OIS-34 = Ontological insecurity scale; O-LIFE = Oxford-Liverpool Inventory of Feelings and Experiences; PAM = Psychosis Attachment Measure; PBI = Parental Bonding Instrument PDI-21 = Peters et al. Delusional Inventory—21 items; PLEs = Psychosis-Like Experiences; PQ = Prodromal Questionnaire; PSS = Perceived Stress Scale; Rotter I-E = Rotter Internal External Locus of Control Scale; RPBS = Revised Paranormal Belief Scale; RQ = Relationship Questionnaire; SCID-I = Structured Clinical Interview for DSM-IV Axis I disorders; SCL-90-R = Symptom Checklist 90-Revised; SIDP-IV = Structured Interview for DSM-IV Personality; SIPS = Structured Interview for Psychosis-risk Syndromes; SNAP = Schedule for Nonadaptive and Adaptive Personality; SPIKE = Structured Psychopathological Interview and Rating of the Social Consequences of Psychological Disturbances for Epidemiology; SPQ = Schizotypal Personality Questionnaire; STA = Schizotypal personality scale; STAI = State-Trait Anxiety Inventory; TAPS-1 = Tobacco, Alcohol, Prescription medication, and other substance use screening scales; TEC = Traumatic Experiences Checklist; TLEQ = Traumatic Life Events Questionnaire; WSS = Wisconsin Schizotypy Scales.

$^*p < .05$

$^{**}p < .01$

$^{***}p < .001$

**Table 4. Numbers of significant (and non-significant) associations between childhood trauma types and dimensions of schizotypy.**

| Childhood Trauma types | Positive domain | Negative domain | Disorganised domain | Total schizotypy |
|---|---|---|---|---|
| Emotional abuse | 20 (9) | 20 (0) | 5 (1) | 10 (0) |
| Physical abuse | 14 (16) | 11 (10) | 5 (2) | 9 (3) |
| Sexual abuse | 11 (21) | 3 (20) | 4 (4) | 10 (4) |
| Neglect | 21 (6) | 10 (5) | 6 (0) | 15 (1) |
| Other | - | - | - | 8 (9) |
| Total trauma scores | 11 (0) | 5 (3) | 1 (0) | 14 (1) |

*Note.* Numbers outside the parentheses refer to significant associations observed in studies and inside the parentheses to non-significant associations; Neglect refers to both physical and emotional neglect; Examples of 'Other' include accidents, disasters, threatening events, family problems, conflicts with and between parents, household mental health difficulties.

effects and the pooled effect across studies. The $I^2$ statistic (which can be derived from the Q) is an intuitive expression of heterogeneity which describes the percentage of variation across studies due to heterogeneity rather than chance. An $I^2$ of 0% was taken to indicate no observed heterogeneity, 25% for low, 50% for moderate and 75% for high heterogeneity [see 125]. All meta-analyses used a random effects model and analysis was conducted using Comprehensive Meta-Analysis (version 2.2).

The pooled effect sizes for the associations between abuse/neglect types and schizotypy are presented in Table 5. Significant associations were found between schizotypal traits and all forms of abuse and neglect, and all effect sizes were highly heterogeneous. The association for emotional abuse was significantly larger than for all other forms of abuse and neglect (physical Q = 15.58, p < .001; sexual Q = 5.27, p = .02; emotional neglect Q = 6.11, p = .01; physical neglect Q = 4.53, p = .03); and no other abuse/neglect effects sizes differed significantly from each other.

## Publication bias

We conducted Trim and Fill analyses for publication bias. Trim and Fill analysis both identifies and corrects for asymmetry in funnel plots that reflects possible publication bias [126] e.g. with smaller studies producing larger outlying effect sizes in one direction. The method 'trims' the smaller studies underpinning the funnel plot asymmetry, the trimmed funnel plot is used to estimate the true 'centre' of the funnel and then replaces the missing studies around the centre. The final estimate of the true mean, and its variance, are then based on the 'filled' funnel plot.

**Table 5. Meta-analysis of correlations between schizotypy and five key forms of childhood abuse and neglect.**

| Type of childhood trauma | k | N | r | 95% CI | Z | Q | $I^2$ |
|---|---|---|---|---|---|---|---|
| Emotional abuse | 11 | 6,702 | .33 | .28 to .37 | 13.95** | 31.14** | 67.88 |
| Physical abuse | 13 | 7, 335 | .20 | .16 to .25 | 8.10** | 44.27** | 72.89 |
| Sexual abuse | 12 | 6,926 | .25 | .17 to .31 | 6.46** | 85.46** | 87.13 |
| Emotional neglect | 7 | 4,331 | .23 | .15 to .30 | 5.98** | 23.84** | 74.83 |
| Physical neglect | 9 | 5,841 | .25 | .19 to .31 | 7.94** | 35.94** | 77.74 |

*Note.* The association between Emotional abuse and schizotypy was significantly greater than for all other forms of abuse and neglect. No other comparisons were significant

**p < .001

Analysis of emotional abuse indicated two possible missing studies, reducing the effect size to .31 (95% CI .27 to .36); for physical abuse, three missing studies reduced the effect size to .17 (95% CI .12 to .22); none were identified for sexual abuse; two were missing for emotional neglect, reducing the effect size to .18 (95% CI .10 to .26); and none were missing for physical neglect. Nonetheless, the numbers of studies are quite small in some analyses and so, the findings on publication bias may be unreliable.

Following Cohen's guidance for interpreting *r* effect sizes (small = .10, med = .30 and large = .50; [127]), the effect size for childhood emotional abuse is in the medium range; while all other trauma types (physical abuse, sexual abuse, physical neglect and emotional neglect in childhood) were in the small effect size range. This suggests that while all types of abuse and neglect in childhood are associated with schizotypy, childhood emotional abuse is more strongly associated; and was significantly more strongly associated with schizotypy scores than any other type of trauma. We also note that heterogeneity was moderate-to large in all analyses.

## Meta-regression

Although there is no definitive minimum number of studies required for meta-regression, we follow the general recommendation of the Cochrane group with at least 6 to 10 studies for a continuous variable [128, see also 129].

Using meta-regression, we examined age and gender (proportion of women participants), study quality, as well as year of publication as moderators of effect sizes (using a Method of Moments meta-regression approach: see Table 6). The mean ages across studies ranged from 18.70 to 44.80 and the mean proportion of women participants ranged from 49.33 to 100%.

Age was a significant moderator of the effect sizes for sexual abuse and emotional neglect. Effect sizes for sexual abuse were larger in younger participants while those for emotional neglect were larger in older participants. Gender (proportion of women participants) moderated the association between physical abuse and schizotypy, with larger associations in studies with more women participants. Finally, year of publication was a significant moderator for emotional and physical neglect, with older studies reporting larger effects for both forms of neglect.

We also looked at a multiple predictor meta-regression model including age and gender to predict the relationship between childhood trauma and schizotypy and the results did not change. Gender continued to moderate the impact of physical abuse on schizotypy (Z = 2,14, p = .03), while age was ns (Z = 0.79, p = .43). Age was a significant predictor (Z = -2.5, p = .01) of the sexual-abuse-schizotypy relationship, while gender was not (Z = -0.81, p = .42). Finally,

**Table 6. Meta-regression of moderators between forms of childhood abuse/neglect and schizotypy.**

| Type of childhood trauma | Age | % women | Study Quality | Publication Year |
|---|---|---|---|---|
| Emotional abuse | -1.62 | 1.63 | -1.29 | 0.29 |
| Physical abuse | -0.12 | 2.02* | -0.11 | 0.18 |
| Sexual abuse | -2.32** | 0.10 | -0.43 | 1.79 |
| Emotional neglect | 2.02* | 1.78 | -1.61 | -4.24*** |
| Physical neglect | -0.97 | 1.28 | -.12 | -2.28** |

*Note.* Numeric values are Z scores

*p ≤ .05

**p = .02

***p < .001

when age and gender were entered neither was a significant predictor of the childhood emotional abuse-schizotypy association.

## Discussion

This systematic review identified 25 studies examining the association of childhood trauma and schizotypy in 15,253 non-clinical individuals. The current meta-analyses are the first to estimate the pooled effect size for the association between various forms of childhood abuse/neglect and schizotypy in subclinical samples and the emerging importance of this area of research is underscored by 17 of the 25 (68%) studies reviewed here being published since the sole previous systematic review by Velikonja, Fisher [18].

Schizotypy scores were significantly associated with all forms of abuse and neglect in a manner that is consistent with a dose-response interpretation. Almost all studies and their analyses report significant positive correlations between schizotypy and various forms of abuse and neglect (emotional abuse 11/11; physical abuse 10/13; sexual abuse 11/12; emotional neglect 7/7; and physical neglect 9/9). The current meta-analyses are however the first to quantify and compare the relative degree of association between these different forms of childhood trauma and schizotypy. Although schizotypy was significantly associated with all forms of abuse and neglect, only the associations with emotional abuse exceed the small range with an $r = .33$ (equivalent to an OR = 3.55). The association of schizotypal traits with emotional abuse was in the moderate range and crucially, significantly larger than the association for all other forms of abuse and neglect (where effect sizes did not differ significantly from each other).

Our assessment indicated that the mean study quality ranking was marginally lower than the suggested cut-off for individual studies to be identified as methodologically robust [see 18]. Hence, most studies failed to meet the suggested cut-off. We also note that study quality has remained relatively low across time, with no signs of any recent increase. Despite this, study quality was not a significant moderator of the relationship between schizotypy and any of form of abuse or neglect. It is of course possible that the quality measure used fails to adequately capture some potentially relevant variability in study quality, although studies did show variability in quality scores (range 4–10 on a scale of 0–14). Certain specific areas of methodological weakness might however be identified. In particular, the vast majority of studies (21/25) failed to control for, assess or comment upon possible confounds–such as controlling for demographic variables, family history, current levels of anxiety or depression and so on.

The studies were well-powered–taking the smallest effect size being 0.2 (for physical abuse) would require 150 participants to detect at 80% power. All studies included in the meta-analyses except two [62, 82] had sufficient sample size (with a median in excess of 400) to detect even the smallest effect size reported here. This power analysis would of course apply to detecting a simple correlation; however, most studies are assessing a range of associations between various types of trauma and various schizotypal traits and any power assessment needs to accommodate multiple testing. The current meta-analysis and effect sizes, hopefully provide some basis for more specific hypothesis testing in future studies.

The stronger association between emotional abuse and schizotypy suggests a more consistent and pervasive link than appears to occur with either sexual or physical abuse. The latter forms of abuse differ from emotional abuse insofar as they might be linked more to individual abusive acts. Emotional abuse is also likely to accompany these other forms of abuse and neglect, and may heighten the effect of other abusive acts [130, 131]. It is also feasible that emotional abuse has a more pervasive influence partly because it might persist undetected for longer. Related to the potentially pervasive and interactive influence of emotional abuse, one limitation of current studies is that they rarely assess the independent contributions of specific

abuse/neglect types and so effect size estimates may be non-independent. Two studies [63, 69] have however attempted to look at the independent impact of trauma types within a regression framework. Goodall, Rush [63] looked at emotional abuse, sexual abuse, physical abuse, emotional neglect and physical neglect as predictors of schizotypy scores (in 283 non-clinical participants) and found that only emotional abuse remained significant when all predictors were entered together. More recently, Toutountzidis, Gale [69] used a similar regression-based approach finding that schizotypy scores were significantly predicted by both emotional and physical abuse, although only emotional abuse remained a significant predictor when gender was entered into the model.

Our meta-analyses identified substantial levels of heterogeneity for all effect sizes, with $I^2$ values ranging from 68–87%. We explored this heterogeneity using meta-regression by focusing on the key participant variables of age and gender as well as study quality and year of publication. Velikonja, Fisher [18] speculated that exploring age and gender differences as possible mediators of the childhood trauma-schizotypy relationship might aid our understanding of the aetiology of psychotic symptoms and psychotic disorders. We found that age was a significant predictor of the sexual abuse-schizotypy association, with stronger associations emerging in younger samples. Age also significantly moderated the emotional neglect-schizotypy relationship, but associations were stronger in older samples. The opposing impact of age in these two types of trauma raises the possibility that additional variables are likely to be relevant. For example, age may impact how people process memories regarding maltreatment in childhood and this may depend upon the type of trauma experienced. It is also possible that temporal changes in social norms are relevant, such that earlier adverse events are not viewed as negatively as they might be now or the likelihood of such events may even have reduced across time.

Turning to gender, the physical abuse-schizotypy link was larger in samples containing more women participants. Given the importance of gender in studies of trauma in clinical psychosis groups, surprising we identified only two non-clinical studies reporting associations separately for men and for women [61, 69]. Reviews have documented gender differences in the prevalence of different traumas reported in psychosis patients [132, 133]; however, the findings are not always consistent and crucially, we should not assume that trauma and schizotypy (or indeed psychosis) shows the same dose-response across men and women. For example, Toutountzidis, Gale (69) found that while men reported significantly more physical trauma experienced in childhood than women, it failed to correlate with schizotypy in men, but did in women. As noted above, the moderating role of gender was also borne out by our meta-regression analyses showing that the physical abuse-schizotypy link was stronger in studies with more women as participants. Turning to the role of age, the link between sexual abuse and schizotypy scores was larger in younger samples and several possible explanations exist. For example, it may be that sexual abuse differentially impacts schizotypy in those who are younger. Another possibility is that older samples are less willing to report sexual abuse (and/ or that younger samples feel safer to disclose on this topic) and this the apparently stronger association in the younger is a confounded by willingness to reveal. The sampling is however somewhat skewed in terms of both age (with a median age of 25) and gender. The ratio of women to men participants in the studies reviewed here is around 2:1; and all studies except one [82] had a majority of women participants. The broadening of the focus onto older and more men as participants is something future studies should address.

Although the overwhelming majority of studies report significant associations between early childhood trauma and schizotypy, a key unanswered question centres on whether any specific adversities are more reliably associated with any specific trait domains or symptoms. This review identifies considerable variation in the types of measures used, but crucially also at

the level of analysis. Some have calculated a total score for early life adversities [70] while more recent studies have tended to separately assess different types of adversity (e.g., physical, emotional, sexual abuse and neglect: e.g., [69, 81, 86]). Similarly, some studies [64, 66, 67, 88] assessed several schizotypy dimensions (e.g., cognitive-perceptual, interpersonal, disorganised), some have broken these domains down further into multiple trait subscales (e.g., 8 subscales: [64]; 9 subscales: [69]), subscales and total scores [e.g., 88] while others have assessed total schizotypy scores only [e.g., 61]. Although variability in the level of analysis is likely to contribute to the high heterogeneity of effect sizes, the data did not permit looking at this in a sub-group analysis. Nonetheless, a simple vote-count suggests that certain relationships seem promising with some invariable associations emerging. The most consistent association being reported for childhood emotional abuse both with *negative* schizotypal traits (e.g., anhedonia, no close friends) (20/20 associations) and with total schizotypy scores (10/10); for physical/ emotional neglect (6/6 studies) and the *disorganised* domain (e.g. odd behaviour and odd speech); and for total trauma scores (11/11 studies) with the *positive* domain (e.g., paranoid ideation, ideas of reference). It is notable that childhood sexual abuse showed inconsistent associations with the various schizotypy domains—being significant in 11/32 assessments for the positive domain; 3/23 for the negative domain; and 4/8 for disorganisation—but did show more reliable association with total schizotypy scores (10/14). We note however that vote-counts do not accommodate sample size differences across studies, statistical power, or the size of effects in studies and not an alternative to meta-analysis [134]. Unfortunately, it was not possible to derive sufficiently consistent levels and types of data to meta-analyse more specific associations between traumas and specific domains or even symptoms.

Little work has thus far focused at the symptom-specific level of analysis. For example, in the case of voice-hearing, only one study [66] has thus far examined the relationship with trauma. Looking at a student sample, Cole and colleagues [66] found significant links between self-reported auditory hallucinations (LSHS) and all types of childhood traumas (sexual, physical, and emotional abuse, and neglect); and for self-reported delusional ideation (PDI-21) with all traumas apart from sexual abuse. This lack of attention is surprising given that so many studies of clinical samples with psychosis have reported links between sexual trauma and voice-hearing and delusions (for a recent meta-analysis, see [135]. The links between trauma and specific symptoms clearly merits further research in non-clinical samples.

## Strengths and limitations

Although a major advantage of this review is the large number of participants and studies, the variety of measurement approaches meant that meta-analyses were limited to broad-brush approaches. Different studies of course have different aims and employ different measurement scales to assess both childhood trauma and schizotypy. In the case of measuring the latter, studies also vary considerably in whether they report total scores, domain (e.g., positive, negative, disorganised) subscale scores, or indeed, employ measures focussed specifically on symptoms (e.g., hallucinations or delusions). In general, studies assessing general population samples have not made a concerted effort to establish the parallels of the same trauma-symptom links highlighted in the clinical literature. For example, clinical studies have tended to emphasise the relationships of trauma types with specific symptoms such as delusions and hallucinations [135]. Future research would benefit from focusing on how childhood trauma might impact specific trait clusters/domains and symptoms in the healthy population. This would then make it more possible to look not just at more specific links, but to make more comparisons with the existing clinical literature.

Turning to other measurement issues, ultimately the strength of reported relationships between trauma and schizotypy depend upon the reliabilities of the measures themselves. Although questions have arisen about the reliability of the self-reporting of childhood adversities in the context of clinical psychosis [e.g., 136], retrospective reports by psychosis patients are stable over time, not especially influenced by current psychopathology, consistent across different assessment instruments and correspond with clinical case notes [e.g., 137]. The most common measure of childhood trauma employed was the Childhood Trauma Questionnaire, [CTQ; 30, 31] being used in 15 of the 25 studies. A recent review [138] of 52 child abuse measurement instruments reported that the CTQ was the most thoroughly investigated and has the strongest levels of evidence overall and strong for: internal consistency, reliability, content validity, structural validity and convergent (hypothesis testing) validity. While we could find no evidence on the reliability of trauma reporting in schizotypy, recall is less likely to be affected by psychosis-like experiences or memory/cognitive deficits; although those with psychometrically defined schizotypy do show relatively subtle cognitive deficits on neuropsychological tests [139]; see also [140] for a recent meta-analysis. We also note that even when objective cognition is intact on testing, schizotypy has also been associated with subjective cognitive problems that may not be detectable on objective tests [141]. So, we cannot exclude the possibility that mild or subjective forms of cognitive impairment, including for memory and executive function, might impact memory recall and reliability in schizotypy.

Six studies used a scale that tests for prodromal attenuated psychosis symptoms [65, 74, 81, 89, 110, 112]. Although these studies did not measure schizotypal traits per se, it is difficult to confidently assert whether they examine trait-like or state-like subclinical symptoms of psychosis for two key reasons. First, the cross-sectional nature of the studies does not permit a test of symptom/trait persistence. Second, all measures use similar questions to assess psychosis-like experiences in non-clinical samples (e.g., Have you had the sense that some person or force is around you, although you couldn't see anyone?; both in PQ and SPQ). Third, only two studies using state measures were included in the meta-analyses [65, 74] and while these state measures revealed correlations that were a little smaller, they did not differ significantly from trait measure effect sizes. So, despite variability in the tests used and the level at which scoring occurs, the pattern of associations across studies and measures was robust.

A clear limitation concerns the failure of most studies to meet the suggested cut-off for robust methodological quality; and that study quality has showed no signs of improving in recent years. While Velikonja, Fisher [18] noted study quality variability in their review, the 17 newer studies added here since that previous review do not evidence any recent increase in quality. We also note that individual quality items assessed here were not weighted and so, any simple comparisons across studies using summed total quality scores should be regarded with some caution [see 142–144]. In this context, we note that an area of specific methodological weakness has been the lack of control for potential confounding variables. For example, studies rarely screened for existing mental health problems, with only 4/25 (16%) screening for previous or current mental health disorders [65, 86, 88, 110] and one screened for possible neurological problems [71].

We also note the lack of diversity in sampling to date. Most studies have sampled exclusively from undergraduate students (16/25: 64%) and as already noted the ratio of women: men being approximately 2:1. The younger age (median 25) is perhaps understandable given the typical age of onset for psychosis-like symptoms–indeed, some studies have used this rationale for excluding participants outside of the 18–35 age range [e.g., 89]. The focus on more highly educated samples limits representativeness much more–certainly with comparison to the educational attainment of those with psychosis. In this context, we were also unable to assess whether ethnicity impacted effect sizes. Only one-in-three studies (9/25: [61, 62, 64, 66,

69, 71, 81, 89, 112]) provided any ethnicity breakdown. All apart from Alemany, Arias [71] were from the UK or USA and these identified a median proportion of white participants at 75%. The role of ethnicity is important here given evidence that the perceived discrimination in early life may be linked to later psychosis-like experiences [145–147]. This means that samples have often focused on largely young, highly-educated women and often lack information on ethnicity.

Of those that we considered to consist of general population samples, some were clearly constrained by their sampling techniques. Amongst the remainder, Berenbaum, Valera (62) "sought respondents with unusual beliefs (e.g. belief in UFOs)" (p. 144); Fekih-Romdhane, Nsibi [82] assessed the unaffected siblings of those diagnosed with schizophrenia; Mongan, Shannon [112] recruited using the online paid participant pool from Mechanical Turk; Gaweda, Goritz [65] also from an online academic participant panel (called WiSO); Powers, Thomas [64] recruited from obstetric/gynaecological clinics; and Rössler, Hengartner [72] recruited a prospective study from amongst men conscripts and women on the electoral register in Zurich from mid 1970s. Finally, Toutountzidis, Gale [69] sampled the general public using an online survey and Startup [120] who used a university voluntary participant panel.

We also note that only three studies screened out participants with any personal history of mental disorder [65, 86, 88], another for psychotic disorders [110] and one screened for a family history of suicide [86]. There were several studies that did not exclude family and personal history of mental health problems [e.g., 20, 63, 66, 67, 74] or whose sampling did not exclude those with self-declared mental health issues [69]. None screened and excluded for a family history of mental disorder. Indeed, family history was an inclusion criterion in one study i.e. the study of unaffected twins of those with schizophrenia [82].

Of course, the conclusion of the current systematic review and meta-analysis are prone to the limitations of correlational research generally. Aside from psychosis-like experiences occurring as a consequence of childhood adversity, the correlational nature of the research does mean that we should acknowledge alternative interpretations. In particular, trauma may result as a consequence of psychosis symptoms or in full-blown psychosis, as a result of involuntary treatment experiences [148, 149]. Despite this, given the subclinical levels of psychosis symptoms in adulthood, it seems less likely that trauma is induced by subclinical psychosis-like experiences earlier in childhood.

Finally, as with psychosis, the links between early life trauma and schizotypy are undoubtedly complex and possible confounding factors cannot be eliminated. For example, it is known that higher levels of parental mental health problems are associated with various forms of abuse. Doidge, Higgins [150] assessed a range of possible child, parent and family risk factors for child maltreatment in a prospective 27-year population-based birth cohort of 2443 Australians. They found that parental mental health problems were associated with high intensity emotional abuse (OR = 2.06), low intensity emotional abuse (OR = 1.90), neglect (OR = 3.60), physical (OR = 2.31) and sexual abuse (OR = 3.33). It is also possible that children living with a parent who has mental health problems might view this as a form of trauma [see 151]. Future studies need to control for parental mental health status.

## Conclusions

This review shows that significant associations exist between childhood trauma and psychometrically defined schizotypy in the non-clinical population. While much research has focused on links between early life trauma and adult psychosis, including a possible causal role [19], the current findings show that trauma in childhood does not necessarily lead to the development of clinical presentations of psychosis. Such early life experiences may however impact

the development and maintenance of potentially dysfunctional experiences and ways of thinking that remain sub-clinical. Despite the limitations of cross-sectional studies in terms of deriving causal inferences, studying childhood adversities that do *not* lead to clinical outcomes is as important as where they do–informing discussion around why some people are more 'resilient' than others.

## Supporting information

**S1 Appendix.**
(DOCX)

**S2 Appendix.**
(DOCX)

## Author Contributions

**Conceptualization:** Diamantis Toutountzidis, Tim M. Gale, Karen Irvine, Shivani Sharma, Keith R. Laws.

**Data curation:** Diamantis Toutountzidis.

**Formal analysis:** Keith R. Laws.

**Methodology:** Diamantis Toutountzidis, Tim M. Gale, Karen Irvine, Shivani Sharma, Keith R. Laws.

**Project administration:** Diamantis Toutountzidis.

**Supervision:** Tim M. Gale, Karen Irvine, Shivani Sharma, Keith R. Laws.

**Visualization:** Diamantis Toutountzidis.

**Writing – original draft:** Diamantis Toutountzidis.

**Writing – review & editing:** Diamantis Toutountzidis, Tim M. Gale, Karen Irvine, Shivani Sharma, Keith R. Laws.

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
