## [Decision Letter · Decision Letter 0]

12 Jan 2022

PONE-D-21-32224Childhood trauma and schizotypy in non-clinical samples: a systematic review and meta-analysisPLOS ONE

Dear Dr. Toutountzidis,

Thank you for submitting your manuscript to PLOS ONE. After careful consideration, we feel that it has merit but does not fully meet PLOS ONE’s publication criteria as it currently stands. Therefore, we invite you to submit a revised version of the manuscript that addresses the points raised during the review process. The reviewers feel that your manuscript is important and relevant, particularly to individuals who study resiliency. They have several comments that they feel would strengthen the manuscript overall.

We look forward to receiving your revised manuscript.

Kind regards,

Sarah Hope Lincoln

Academic Editor

PLOS ONE

Journal Requirements:

“Part of the review was funded by a PhD bursary from the University of Hertfordshire to DT. The funder did not have any involvement in study design; in the collection, analysis and interpretation of data; in the writing of the report; and in the decision to submit the article for publication.”

We note that you have provided funding information within the Funding Section. Please note that funding information should not appear in the Acknowledgments section or other areas of your manuscript. We will only publish funding information present in the Funding Statement section of the online submission form.

“Part of the review was funded by a PhD bursary from the University of Hertfordshire to DT. The funder did not have any involvement in study design; in the collection, analysis and interpretation of data; in the writing of the report; and in the decision to submit the article for publication.”

Reviewers' comments:

Reviewer's Responses to Questions

**Comments to the Author**

1. Is the manuscript technically sound, and do the data support the conclusions?

Reviewer #1: Yes

Reviewer #2: Yes

2. Has the statistical analysis been performed appropriately and rigorously? 

Reviewer #1: Yes

Reviewer #2: Yes

3. Have the authors made all data underlying the findings in their manuscript fully available?

Reviewer #1: Yes

Reviewer #2: Yes

4. Is the manuscript presented in an intelligible fashion and written in standard English?

Reviewer #1: Yes

Reviewer #2: Yes

5. Review Comments to the Author

Reviewer #1: The primary aim of this review paper is to present the association between psychometric schizotypy (i.e., schizotypal traits in non-clinical samples) and early life adversities. The manuscript is well-written, provides quantitative summaries of available literature according to current best practices, and results are clearly presented. The topic is important and contributes to an area that is thus far under researched. Recommendations are made to improve presentation of the data extraction process and interpretation of the results and some minor recommendations are also offered:

1. Introduction: The introduction is well-written and nicely reviews the rationale/existing support for childhood adverse experiences as risk factors for psychosis-like experiences (PLEs) and schizotypy. Given this is a review of correlational observations, what evidence exists, if any, for schizotypy or PLEs to be considered risk factors for experiencing adverse childhood experiences or for increasing the likelihood of experiencing more adverse childhood experiences. For example, is there longitudinal research that establishes PLEs or schizotypy (may be difficult because typically understood as a diagnosis in adulthood) preceding adverse childhood events?

2. Introduction: How is “childhood adverse event” defined, any events prior to age 18? Please clarify.

3. Method: Many steps of the methods are clearly described in detail which is important for replication. One additional clarification How were articles screened at the title, abstract, and full-text level? Did two independent reviewers screen all articles at all levels of review (i.e., title, abstract, and full-text level) and resolve discrepancies? Was double-screening completed at just at an abstract or full-text level, please clarify lines 148-149 (“screening and eligibility assessment…).

4. Method: Since the study was not preregistered (which the authors discuss in the introduction), it would be helpful to have more information about structure of coding, training of coders, and a priori protocols put in place prior to data extraction. For example, how were coders trained? Were there ongoing consensus meetings, did multiple coders extract data from a same group of articles and discuss discrepancies before completing all extraction? Was there a manual or other documentation of coding decisions/definition of categories made available? The same question would be helpful to address for the screening process, did all coders independently screen all potential articles and then resolve discrepancies or was there a smaller subset of screening done and then some type of reconciliation that happened to make sure coders were employing similar inclusion criteria? Relatedly, were there categories that were coded for (e.g., race/ethnicity) that were not included in the final analyses due to insufficient data? It is common to see meta-analyses discuss categories that were planned but not able to be included due to insufficient study reporting. This can be an important future direction to discuss in the manuscript (e.g., if only 5% of studies reported on participant race and ethnicity, the authors can provide citable statements to this effect and recommend that future studies of psychometric schizotypy include this information).

5. Method: please provide more detailed information regarding how “general population sample” was defined. The authors specify that no clinical psychosis or personality disorder could be present but do not specify if study participants could have other mental health diagnoses. If participants could have other mental health diagnoses (e.g., depression) or be part of a treatment-involved sample (e.g., outpatient clinic participants) it would be helpful to have this information included in Table 3 of study characteristics (e.g., inclusion of mental health diagnoses Y/N). If study participants were relatives of individuals with schizotypy or psychosis or otherwise identified as high-risk for schizotypy, were the included in this meta-analysis? If yes, it would be helpful to present this as well in Table 3.

a. If family history was reported in studies at a sufficient rate, a recommendation is made to include family history as a predictor of effect size as well.

6. Results: If possible, a recommendation is made to present the country (or perhaps reduce by broader area/continent) for each study in Table 3 and to include country as a predictor of effect sizes in the analyses.

7. Discussion: If possible, a strong recommendation is made to review included articles for reporting on race. If studies report race of study participants at an acceptable level (e.g., even being able to dichotomize % white, although no ideal), it would be important to enter race as a predictor of effect size. A recommendation is made to include a discussion of how race impacts adverse childhood experiences and the relationship to PLEs/schizotypy. There is a growing body of work on how race impacts this relationship and may be important to better understand the findings of this study (e.g., Anglin et al., 2014; Gibson et al., 2016)

a. References:

i. Anglin DM, Lighty Q, Greenspoon M, Ellman LM. Racial discrimination is associated with distressing subthreshold positive psychotic symptoms among US urban ethnic minority young adults. Soc Psychiatry Psychiatr Epidemiol. 2014 Oct;49(10):1545-55. doi: 10.1007/s00127-014-0870-8. Epub 2014 Apr 3. PMID: 24695907.

ii. Gibson, L. E., Alloy, L. B., & Ellman, L. M. (2016). Trauma and the psychosis spectrum: A review of symptom specificity and explanatory mechanisms. Clinical Psychology Review, 49, 92-105.

Minor points:

- Abstract: lines 22-23, edit suggested “By contrast, no systematic review or meta-analysis has…”

- Some statistical reporting is inconsistent (e.g., italicizing all rs relating to effect size throughout).

- Use gender/sex consistently throughout manuscript with fidelity to how these were assessed by study to adhere to language free of bias. For example, lines 461-462 (“Turning to gender, the physical abuse-schizotypy link was larger in samples containing more female participants”). Use of sex (e.g., male, female) is recommended when referring to sex assigned at birth while gender refers to a social construct/identify (e.g., man, woman). If assessing/analyzing sex, a recommendation is made to define as “sex assigned at birth” and then use “male/female” throughout the manuscript rather than conflate sex and gender terms (e.g., https://apastyle.apa.org/style-grammar-guidelines/bias-free-language/gender).

Reviewer #2: The current study assessed whether early life adversities are associated with schizotypal personality traits in the non-clinical population. The authors conducted a systematic review and meta-analysis of associations between early life adversities schizotypal traits in non- clinical samples. Twenty-six studies (N=15,818 participants) were included in the review. Meta-analyses showed that all forms of abuse (emotional, physical and sexual) and neglect (emotional and physical) were significantly associated with schizotypy. The association of schizotypy traits with emotional abuse (r = .33: 95%CI .30 to .37) was significantly larger than for all other form of abuse or neglect. The current study found a dose-response relationship between all forms of abuse/neglect and schizotypy scores in non-clinical samples with the stronger association being for emotional abuse.

The manuscript addressed an interesting question regarding early childhood abuse and its relationship to personality traits related to schizophrenia spectrum disorders in a nonclinical sample using metanalysis. Given that early abuse is often associated with psychosis, this research is of interest to those who study resiliency. Some clarifications are needed to make the article more readable.

Introduction

• Page 5, line 72: Provide examples of schizotypal traits.

Results

• Page 10, line 190:Why were studies that scored below 8 included?

• Page 41: please explain the Trim and fill analysis method.

Discussion

• Page 51: “On quality assessment, fewer than half of the studies were rated as being methodologically robust” This should be included in your weaknesses section.

• Page 5, line 79: “Adversities in early life are common, with approximately 40% of the general adult population reporting at least one type of adverse experience (e.g. parental mental illness, domestic violence, physical, emotional, and sexual abuse, neglect) before the age of 18;” Can you check ACE scale to see what other adverse experiences might be applied? Maybe include parental substance abuse or parents in jail as factors.

• Page 6-7, ending at line 119: Unsure of what the numbers 1 and 2 mean here; are there any combinations of these search terms in 1 and then in 2?

• Page 7, line 137: “The review followed PRISMA guidelines, however, it was not pre-registered;” Please summarize the PRISMA guidelines

• Page 8, line 143: “. . . (c) include a measure of childhood trauma before the age of 18 years (excluding peer victimization; i.e., bullying; to assess for trauma where the perpetrator is an adult;” Peer victimization/bullying was included on page 6 though

• Page 11, line 202: “The mean age of participants across the 26 studies was 27.06 with a standard deviation of 12.21 (age range 18-95); median = 25;” Why only assess such a young population? Please acknowledge this as a limitation in the discussion

• Page 11, line 216: “The CATS, CECA and ACE-Q assess all aforementioned types of adversity as well as other adversities, such as parental conflict and control;” Be more specific such as parental mental illness, substance abuse, family member going to prison

• Page 17, lines 277-279: “. . . found that physical abuse showed links with an array of schizotypal traits only in women, that emotional abuse associated with an array of schizotypal traits in both sexes and that sexual abuse did not have a link with schizotypy” Should be in an earlier paragraph

• Page 39, Table 4: Specify are these positive and negative affects? Or something else?

o Put “Child” before each type of abuse

o “Neglect” should be split into both Physical Neglect and Emotional Neglect separately

• Page 39, lines 314-316: “Emotional abuse invariably correlated with the negative dimension and with total schizotypy scores; Neglect (physical and emotional) with the disorganized dimension; and finally, total trauma scores with the dositive dimension scores;” What are negative, disorganized, and positive dimensions? Please specify

• Page 38, line 320: Specify these measures as emotional abuse, physical abuse, and sexual abuse

• Page 40, line 330: “Heterogeneity and variance among effect sizes of studies were assessed using the q statistic and the I2 statistic;” Some explanation of Q statistics and I2 statistics will be helpful

• Page 41, Table 5: Specify each measure as Child Emotional Abuse, Child Physical Abuse, Child Sexual Abuse, Child Emotional Neglect, and Child Physical Neglect.

• Page 42, lines 355-359: “. . . the effect size for (child) emotional trauma is in the medium range; while all other trauma types (physical abuse, sexual abuse, physical neglect and emotional neglect) (in childhood) were in the small effect size range. This suggests that while all types of abuse and neglect are associated with schizotypy, (childhood) emotional abuse is more strongly associated; and was significantly more strongly associated with schizotypy scores than any other (childhood) trauma or neglect;” Specify these traumas and abuse as occurring in childhood

• Page 43, line 379: “We also looked at a multiple regression model including both age and gender as predictors and the results did not change;” What are the outcome variables? What other predictors are there?

• Page 43, line 380: “Gender continued to moderate the impact of physical abuse on schizotypy, while age was ns;” Provide more detailed explanation on how moderation was established.

• Page 48, line 497: “The most consistent being reported for (childhood) emotional abuse and the domain of negative schizotypal traits and total schizotypy . . .” Add childhood to specify emotional abuse.

• Page 48, lines 498-502: “for physical/emotional neglect and the disorganized domain; and for total trauma scores with the positive domain. It is notable that early sexual trauma had little evidence of associations with any specific schizotypy domains but did show links with total schizotypy;” This information is really important but hard to understand. Please refresh readers on what disorganized and positive domains mean.

• Pages 48-49, lines 507-508: “. . . all types of traumas (sexual, physical, and emotional abuse, and neglect) . . .” Are these all childhood variables? If so, please specify.

• Page 51, line 558: “Finally, as with psychosis, the links between early life trauma and schizotypy are undoubtedly complex and possible confounding factors cannot be eliminated;” Please acknowledge that the sample is very young (mean age = 26)

6. PLOS authors have the option to publish the peer review history of their article (what does this mean?). If published, this will include your full peer review and any attached files.

Reviewer #1: No

Reviewer #2: **Yes: **Weili Lu

---

## [Author Response · Author response to Decision Letter 0]

4 Mar 2022

Reviewer #1: Thank you for your helpful suggestions. We have incorporated them into our revision and we provided further comments below.

1. Introduction: The introduction is well-written and nicely reviews the rationale/existing support for childhood adverse experiences as risk factors for psychosis-like experiences (PLEs) and schizotypy. Given this is a review of correlational observations, what evidence exists, if any, for schizotypy or PLEs to be considered risk factors for experiencing adverse childhood experiences or for increasing the likelihood of experiencing more adverse childhood experiences. For example, is there longitudinal research that establishes PLEs or schizotypy (may be difficult because typically understood as a diagnosis in adulthood) preceding adverse childhood events?

We have acknowledged this important point by adding the following paragraph and references in the discussion (p.55, line, 623-630)

“Of course, the conclusion of the current systematic review and meta-analysis are prone to the limitations of correlational research generally. Aside from psychosis-like experiences occurring as a consequence of childhood adversity, the correlational nature of the research does mean that we should acknowledge alternative interpretations. In particular, trauma may result as a consequence of psychosis symptoms or in full-blown psychosis, as a result of involuntary treatment experiences (148, 149 ). Despite this, given the subclinical levels of psychosis symptoms in adulthood, it seems less likely that trauma is induced by subclinical psychosis-like experiences earlier in childhood.”

2. Introduction: How is “childhood adverse event” defined, any events prior to age 18? Please clarify.

We have clarified this - in the introduction - by adding that it does mean before the age of 18 (p.3, line 61)

3. Method: Many steps of the methods are clearly described in detail which is important for replication. One additional clarification How were articles screened at the title, abstract, and full-text level? Did two independent reviewers screen all articles at all levels of review (i.e., title, abstract, and full-text level) and resolve discrepancies? Was double-screening completed at just at an abstract or full-text level, please clarify lines 148-149 (“screening and eligibility assessment…).

Two independent reviewers (DT and KRL) screened all papers at title and abstract phase – we have added this detail to the paper. Both (DT and KRL) also extracted the data for the meta-analyses (now p.6, lines 118-119; p. 7, 145-146)

4. Method: Since the study was not preregistered (which the authors discuss in the introduction), it would be helpful to have more information about structure of coding, training of coders, and a priori protocols put in place prior to data extraction. For example, how were coders trained? Were there ongoing consensus meetings, did multiple coders extract data from a same group of articles and discuss discrepancies before completing all extraction? Was there a manual or other documentation of coding decisions/definition of categories made available? The same question would be helpful to address for the screening process, did all coders independently screen all potential articles and then resolve discrepancies or was there a smaller subset of screening done and then some type of reconciliation that happened to make sure coders were employing similar inclusion criteria? Relatedly, were there categories that were coded for (e.g., race/ethnicity) that were not included in the final analyses due to insufficient data? It is common to see meta-analyses discuss categories that were planned but not able to be included due to insufficient study reporting. This can be an important future direction to discuss in the manuscript (e.g., if only 5% of studies reported on participant race and ethnicity, the authors can provide citable statements to this effect and recommend that future studies of psychometric schizotypy include this information).

Coding was quite simple for data extraction – only involving two authors (DT and KRL) independently extracting the correlational values and sample sizes; and the data for various moderators (age, percentage of females and year of study)

We have added additional text to describe data extraction “All relevant data for calculating effect sizes (correlation values, sample sizes) and moderator variables (age, percentage of female participants and year of publication) were extracted independently by two authors (DT and KRL).” (p.8, lines 175-177)

On the second point regarding ethnicity/race, we were unable to include any direct analysis on race/ethnicity as details were reported in only a third of studies (9/25: 61, 62, 64, 66, 69, 71, 81, 89, 112) provided a breakdown (with a proportion of white participants ranging from 5.6 to 100%). Only one study reported a minority of white participants (64). The median proportion of white participants of these 9 studies was 75% white. All of these studies (apart from [71] were from the UK or USA). 

We have also added details on this to the discussion with the recommendation of more consistent reporting of such important background details. (p.53-54, lines 598-604)

5. Method: please provide more detailed information regarding how “general population sample” was defined. The authors specify that no clinical psychosis or personality disorder could be present but do not specify if study participants could have other mental health diagnoses. If participants could have other mental health diagnoses (e.g., depression) or be part of a treatment-involved sample (e.g., outpatient clinic participants) it would be helpful to have this information included in Table 3 of study characteristics (e.g., inclusion of mental health diagnoses Y/N). If study participants were relatives of individuals with schizotypy or psychosis or otherwise identified as high-risk for schizotypy, were the included in this meta-analysis? If yes, it would be helpful to present this as well in Table 3.

a. If family history was reported in studies at a sufficient rate, a recommendation is made to include family history as a predictor of effect size as well.

General population samples here means that they did not have any diagnosis of clinical psychosis or personality disorder. Details about other mental health issues such as anxiety and depression were rarely reported and so, could not be excluded.

We have clarified that while psychosis was an exclusion criterion, participants could have included members of the general population with other disorders (as these were not screened by most studies) – and we have also added this as a discussion point i.e. the lack of screening for other mental health issues. We also have clarified some issues relating to other mental health problems in the discussion (p.54, lines 616-622)

6. Results: If possible, a recommendation is made to present the country (or perhaps reduce by broader area/continent) for each study in Table 3 and to include country as a predictor of effect sizes in the analyses.

We have added information on location of studies – second line of results section (p. 8-9, line 188-189)

“The studies originated in the following national locations: Europe =14; USA =6; Australia =2; Africa =2; and China =1” 

We could not use country as a subgroup analysis of effect sizes – because any distinction could have been categorical; and probably US and Europe versus elsewhere; however, the latter only have at best 5 studies. Also, the distinction would be somewhat arbitrary e.g. what do China, Africa and Australia have in common than US and Europe?

7. Discussion: If possible, a strong recommendation is made to review included articles for reporting on race. If studies report race of study participants at an acceptable level (e.g., even being able to dichotomize % white, although no ideal), it would be important to enter race as a predictor of effect size. A recommendation is made to include a discussion of how race impacts adverse childhood experiences and the relationship to PLEs/schizotypy. There is a growing body of work on how race impacts this relationship and may be important to better understand the findings of this study (e.g., Anglin et al., 2014; Gibson et al., 2016)

a. References:

i. Anglin DM, Lighty Q, Greenspoon M, Ellman LM. Racial discrimination is associated with distressing subthreshold positive psychotic symptoms among US urban ethnic minority young adults. Soc Psychiatry Psychiatr Epidemiol. 2014 Oct;49(10):1545-55. doi: 10.1007/s00127-014-0870-8. Epub 2014 Apr 3. PMID: 24695907.

ii. Gibson, L. E., Alloy, L. B., & Ellman, L. M. (2016). Trauma and the psychosis spectrum: A review of symptom specificity and explanatory mechanisms. Clinical Psychology Review, 49, 92-105.

We have added a comment upon this in the limitations of the discussion – linking the lack of reported ethnicity data and the links between the experience of discrimination/racism and psychosis-like experiences, and added that suggested references (p. 54, lines 601-604)

Minor points:

- Abstract: lines 22-23, edit suggested “By contrast, no systematic review or meta-analysis has…”

Changed

- Some statistical reporting is inconsistent (e.g., italicizing all rs relating to effect size throughout).

Changed

- Use gender/sex consistently throughout manuscript with fidelity to how these were assessed by study to adhere to language free of bias. For example, lines 461-462 (“Turning to gender, the physical abuse-schizotypy link was larger in samples containing more female participants”). Use of sex (e.g., male, female) is recommended when referring to sex assigned at birth while gender refers to a social construct/identify (e.g., man, woman). If assessing/analyzing sex, a recommendation is made to define as “sex assigned at birth” and then use “male/female” throughout the manuscript rather than conflate sex and gender terms (e.g., https://apastyle.apa.org/style-grammar-guidelines/bias-free-language/gender).

The original papers themselves vary, with some identifying gender and some sex. As we cannot assume sex assigned at birth (from published studies), and most studies have used self-report on this factor, we have adopted a gender-based distinction throughout (women and men).

Reviewer #2: Thank you for your helpful suggestions. We have incorporated them into our revision and we provided further comments below. 

Introduction

• Page 5, line 72: Provide examples of schizotypal traits.

Done (now p.3, lines 50-56)

Results

• Page 10, line 190: Why were studies that scored below 8 included?

We included all studies to highlight the full range of study quality and the fact that study quality needs to be improved; and study quality did significantly predict the heterogeneity in effect sizes.

• Page 41: please explain the Trim and fill analysis method.

We have added the following to the text (p. 43, lines 352-358)

“Trim and Fill analysis both identifies and corrects for asymmetry in funnel plots that reflects possible publication bias (Duval & Tweedie 2000) e.g. with smaller studies producing larger outlying effect sizes in one direction. The method ‘trims’ the smaller studies underpinning the funnel plot asymmetry, the trimmed funnel plot is used to estimate the true ‘centre’ of the funnel and then replaces the missing studies around the centre. The final estimate of the true mean, and its variance, are then based on the ‘filled’ funnel plot”

Discussion

• Page 51: “On quality assessment, fewer than half of the studies were rated as being methodologically robust” This should be included in your weaknesses section.

We have added this to the limitations section (now p. 53, 580-590)

• Page 5, line 79: “Adversities in early life are common, with approximately 40% of the general adult population reporting at least one type of adverse experience (e.g. parental mental illness, domestic violence, physical, emotional, and sexual abuse, neglect) before the age of 18;” Can you check ACE scale to see what other adverse experiences might be applied? Maybe include parental substance abuse or parents in jail as factors.

Examples here were taken from the study cited in the text. It is also true that some scales assess other traumatic experiences, though these were not sufficiently assessed for us to examine them e.g. only one study used the ACE. There was also discussion re. parental mental health and associations with various forms of abuse (p. 55, 632-640)

• Page 6-7, ending at line 119: Unsure of what the numbers 1 and 2 mean here; are there any combinations of these search terms in 1 and then in 2?

The numbers just separated the AND function here – as the numbers seem confusing, we have removed them (comment related to the search terms, now p. 5, lines 100-107)

• Page 7, line 137: “The review followed PRISMA guidelines, however, it was not pre-registered;” Please summarize the PRISMA guidelines

We have added more details on PRISMA requirements (lines 131-134) and the following text

“A PRISMA 2020 compliant flow diagram tool (27) was used to provide record numbers of each stage and reasons for reports exclusion in the full review stage (Fig 1) and the checklist documents if and where relevant information may be located in the paper.”

• Page 8, line 143: “. . . (c) include a measure of childhood trauma before the age of 18 years (excluding peer victimization; i.e., bullying; to assess for trauma where the perpetrator is an adult;” Peer victimization/bullying was included on page 6 though

This is true – we included bullying (and variants) in the search terms to be comprehensive (e.g. in case anything could be extracted from papers that were not identified by the other search terms), but we did not include bullying in the review, because (a) we were interested in assessing trauma were the perpetrator is an adult and (b) there is a line of research that examines links between bullying and psychosis that was beyond the scope of the current review.

• Page 11, line 202: “The mean age of participants across the 26 studies was 27.06 with a standard deviation of 12.21 (age range 18-95); median = 25;” Why only assess such a young population? Please acknowledge this as a limitation in the discussion

Yes, this is a limitation of the original studies, which tend to focus on younger samples (including undergraduate students) – we have expanded the discussion to detail this point.

In this context, we have expanded the limitations section to contain various sampling limitations relating to age, gender, education and ethnicity.

• Page 11, line 216: “The CATS, CECA and ACE-Q assess all aforementioned types of adversity as well as other adversities, such as parental conflict and control;” Be more specific such as parental mental illness, substance abuse, family member going to prison

Added as recommended (p.10, lines 217-218)

• Page 17, lines 277-279: “. . . found that physical abuse showed links with an array of schizotypal traits only in women, that emotional abuse associated with an array of schizotypal traits in both sexes and that sexual abuse did not have a link with schizotypy” Should be in an earlier paragraph

Moved (now p. 14-15, lines 281-283)

• Page 39, Table 4: Specify are these positive and negative affects? Or something else?

o Put “Child” before each type of abuse – 

We have highlighted these details on the column of trauma types to clarify in all tables.

We also added further clarification at the end of the introduction to remind the reader that all references to abuse and neglect are to experiences in childhood and up to the age of 18.

o “Neglect” should be split into both Physical Neglect and Emotional Neglect separately

Some studies provided results for emotional neglect others for physical neglect and some for both physical and emotional neglect together – thus, we presented results for physical and emotional neglect together to avoid adding three separate neglect types i.e. emotional, physical and both in the table. We specify that neglect refers to both emotional and physical neglect in the note below the table. 

• Page 39, lines 314-316: “Emotional abuse invariably correlated with the negative dimension and with total schizotypy scores; Neglect (physical and emotional) with the disorganized dimension; and finally, total trauma scores with the positive dimension scores;” What are negative, disorganized, and positive dimensions? Please specify

We have now expanded upon this in the introduction (see p.3 lines 50-56)

“Schizotypal traits are often classified into three domains that correspond to the key symptom areas of schizophrenia: the positive domain typically incorporates traits that relate to anomalies of cognition (e.g., paranoid ideation, ideas of reference); the negative domain includes interpersonal, emotional and deficit traits (e.g., anhedonia, no close friends); and the disorganisation domain includes traits related to disruptions in the ability to organise and express thoughts and behaviour (e.g. odd behaviour and odd speech)”

Also, there was a description of the multidimensional nature of schizotypy in the text (highlighted on p. 40. lines 303-308.

• Page 38, line 320: Specify these measures as emotional abuse, physical abuse, and sexual abuse

Added (now p.41, lines 323-324)

• Page 40, line 330: “Heterogeneity and variance among effect sizes of studies were assessed using the q statistic and the I2 statistic;” Some explanation of Q statistics and I2 statistics will be helpful

Added (p.41, lines 332-337)

• Page 41, Table 5: Specify each measure as Child Emotional Abuse, Child Physical Abuse, Child Sexual Abuse, Child Emotional Neglect, and Child Physical Neglect.

Done by specifying on column above all trauma types in Table 5

• Page 42, lines 355-359: “. . . the effect size for (child) emotional trauma is in the medium range; while all other trauma types (physical abuse, sexual abuse, physical neglect and emotional neglect) (in childhood) were in the small effect size range. This suggests that while all types of abuse and neglect are associated with schizotypy, (childhood) emotional abuse is more strongly associated; and was significantly more strongly associated with schizotypy scores than any other (childhood) trauma or neglect;” Specify these traumas and abuse as occurring in childhood

Added (now p. 43, lines 367-370)

It has also been specified that the review examines types of trauma experienced in childhood and up to the age of 18; however, we feel it is important for the reader to be reminded throughout the text.

• Page 43, line 379: “We also looked at a multiple regression model including both age and gender as predictors and the results did not change;” What are the outcome variables? What other predictors are there?

We only pitted age and gender against each other as predictors of the trauma-schizotypy links using meta-regression. There were too few variables that had sufficient data points to include additional predictors.

• Page 43, line 380: “Gender continued to moderate the impact of physical abuse on schizotypy, while age was ns;” Provide more detailed explanation on how moderation was established.

This was examined using multiple meta-regression with age and gender as predictors of the trauma-schizotypy associations. Information on how meta-regression was run is provided in meta-regression section of the manuscript (p.44, lines 374-397).

• Page 48, line 497: “The most consistent being reported for (childhood) emotional abuse and the domain of negative schizotypal traits and total schizotypy . . .” Add childhood to specify emotional abuse.

Added (now p. 50) line 510-511) and comment noted as mentioned above re. reminding readers that all abuse and neglect refers to childhood experiences

• Page 48, lines 498-502: “for physical/emotional neglect and the disorganized domain; and for total trauma scores with the positive domain. It is notable that early sexual trauma had little evidence of associations with any specific schizotypy domains but did show links with total schizotypy;” This information is really important but hard to understand. Please refresh readers on what disorganized and positive domains mean.

We have elaborated all schizotypy domains in the introduction (p. 3, lines 50-56) and reminded readers here (p. 50, lines 511-514).

• Pages 48-49, lines 507-508: “. . . all types of traumas (sexual, physical, and emotional abuse, and neglect) . . .” Are these all childhood variables? If so, please specify.

Yes, added

We also specified that all references to abuse and neglect are to childhood experiences in the introduction (page 6, lines 111-113)

• Page 51, line 558: “Finally, as with psychosis, the links between early life trauma and schizotypy are undoubtedly complex and possible confounding factors cannot be eliminated;” Please acknowledge that the sample is very young (mean age = 26)

We have acknowledged the youngish age of the sample and incorporated this into the interpretation of our meta-regression results (p. 44 lines 391- 397, p. 49, 485-492) and as a limitation of the sampling in the discussion.

---

## [Decision Letter · Decision Letter 1]

13 Jun 2022

Childhood trauma and schizotypy in non-clinical samples: a systematic review and meta-analysis

PONE-D-21-32224R1

Dear Dr. Toutountzidis,

We’re pleased to inform you that your manuscript has been judged scientifically suitable for publication and will be formally accepted for publication once it meets all outstanding technical requirements.

Kind regards,

Sarah Hope Lincoln

Academic Editor

PLOS ONE

Additional Editor Comments (optional):

Reviewers' comments:

Reviewer's Responses to Questions

**Comments to the Author**

1. If the authors have adequately addressed your comments raised in a previous round of review and you feel that this manuscript is now acceptable for publication, you may indicate that here to bypass the “Comments to the Author” section, enter your conflict of interest statement in the “Confidential to Editor” section, and submit your "Accept" recommendation.

Reviewer #1: All comments have been addressed

2. Is the manuscript technically sound, and do the data support the conclusions?

Reviewer #1: Yes

3. Has the statistical analysis been performed appropriately and rigorously? 

Reviewer #1: Yes

4. Have the authors made all data underlying the findings in their manuscript fully available?

Reviewer #1: Yes

5. Is the manuscript presented in an intelligible fashion and written in standard English?

Reviewer #1: Yes

6. Review Comments to the Author

Reviewer #1: All previous comments have been addressed by the authorship team. I have no additional comments/concerns for the authorship team.

7. PLOS authors have the option to publish the peer review history of their article (what does this mean?). If published, this will include your full peer review and any attached files.

Reviewer #1: No

---

## [Editor Report · Acceptance letter]

17 Jun 2022

PONE-D-21-32224R1 

Childhood trauma and schizotypy in non-clinical samples: a systematic review and meta-analysis 

Dear Dr. Toutountzidis:

I'm pleased to inform you that your manuscript has been deemed suitable for publication in PLOS ONE. Congratulations! Your manuscript is now with our production department. 

Kind regards, 

on behalf of

Dr. Sarah Hope Lincoln 

Academic Editor

PLOS ONE